# Preparation of a benziodazole-type iodine(III) compound and its application as a nitrating reagent for synthesis of furazans via a copper-catalyzed cascade process
Zhifang Yang, Jun Xu, Yuli Sun, Xuemin Li, Bohan Jia & Yunfei Du ✉

The existing hypervalent I(III) reagents bearing $ONO_2$ group are limited in types and their applications primarily focused on the nitrooxylation reactions featuring a fully-*exo* fashion. Herein, a benziodazole-type $O_2NO$-I(III) compound was prepared and its reaction with β-monosubstituted enamines in the presence of CuI could trigger a radical nitration/cyclization/dehydration cascade to provide a series of less explored but biologically interesting furazan heterocycles. Mechanistically, the benziodazole-type $O_2NO$-I(III) compound acts as a nitrating reagent and incorporates its NO moiety into the final furazan product in a fully-*endo* model, a process of which was proposed to involve nitration, cyclization and dehydration.

In the past several decades, the development and application of hypervalent iodine(III) reagents have received considerable attention from organic chemists for their excellent properties[1–6], including thermodynamic stability, environmental friendliness, and versatile reactivity[7–24]. In contrast to the most well-developed functional-group-transferring transformations enabled by trifluoromethyl-, fluoro-, azido-, alkynyl-, alkenyl-containing hypervalent iodine(III) reagents[7–16], the application of the nitrooxyl ($O_2NO$)-containing I(III) reagents has remained a challenge for organic chemists, as the existing $O_2NO$-I(III) reagents **1a–c** had not found wide application in organic synthesis since their discovery several decades ago (Fig. 1a)[25,26]. It was not until 2020 that Katayev's group realized the first application of $O_2NO$-I(III) reagent **1c** in the preparation of nitrooxylated β-keto esters, 1,3-diketones, malonates, and oxindoles in the absence of oxidants or bases (Fig. 1b)[27]. It is worth mentioning that the asymmetric version of this transformation between the reaction of $O_2NO$-I(III) reagent **1c** with β-keto esters and β-keto amides were further investigated by Deng[28] and Feng's groups (Fig. 1b)[29], respectively. In 2023, Deng's group further accomplished the nitrooxylation of diverse substrates including cyclopropyl silyl ethers, β-keto esters, β-keto amides, 1,3-diketones, and β-naphthol, by using noncyclic $O_2NO$-I(III) compound **1a** as nitrooxylating reagent (Fig. 1b)[30]. In addition, a catalyst-free intermolecular dearomatization reaction of β-naphthols with reagent **1c** under mild conditions to access

various nitrooxylated β-naphthalenones was uncovered by You's group recently (Fig. 1b)[31]. Obviously, the existing hypervalent iodine(III) reagents bearing the $ONO_2$ group are limited in types, and their applications primarily focused on the nitrooxylation reactions featuring a fully-*exo* fashion. In this regard, the development of hypervalent $O_2NO$-containing iodine(III) reagents and searching for their other unique applications should be highly desirable.

Furazans (1,2,5-oxadiazoles)[32–36] constitute an important class of heterocycles that have been applied as energetic materials[37–43] and biologically active agents[44–48]. Accordingly, a great deal of effort has been devoted to the assemblage of this unique class of skeletons. However, the known strategies for accessing Furazans are relatively limited. Literature survey showed that the synthesis of furazans could be realized via deoxygenation of furoxans by tri-substituted phosphite (Fig. 1c, react. 1)[49–51], cyclization of vinyl azides with $NOBF_4$ (Fig. 1c, react. 2)[52], and dehydrative cyclization of vicinal bisoximes (Fig. 1c, react. 3)[34–36,53–63] mediated by alkalinous[34,53] or acidic additives[54–61], $I_2P_4$[62] or $PPh_3$/DIAD[63]. It is worth noting that this last strategy is the most widely-used methodology as vicinal bisoximes, the precursor of furazans, could be readily obtained from hydroxylamination of ammonia with glyoxals, glyoxal monooximes, cyano oximes, or acyl cyanides (Fig. 1c, react. 4)[34,64]. Additionally, Kwong's group recently developed an expeditious metal-free [2 + 2 + 1] radical tandem cyclization reaction of arylketimine,

Tianjin Key Laboratory for Modern Drug Delivery & High-Efficiency, School of Pharmaceutical Science and Technology, Tianjin University, Tianjin, 300072, China.
✉e-mail: duyunfeier@tju.edu.cn

**Fig. 1 | Existing applications of O₂NO-I(III) compounds and known accesses to furazans.** a Known O₂NO-I(III) compounds **1a–c**. b Reported works utilizing O₂NO-I(III) reagents **1a** and **1c** in a fully-*exo* fashion. c Reported strategies for the preparation of furazan derivatives. d This work: a O₂NO-I(III) compound **1d** and its application for synthesis of furazans as a nitrating reagent.

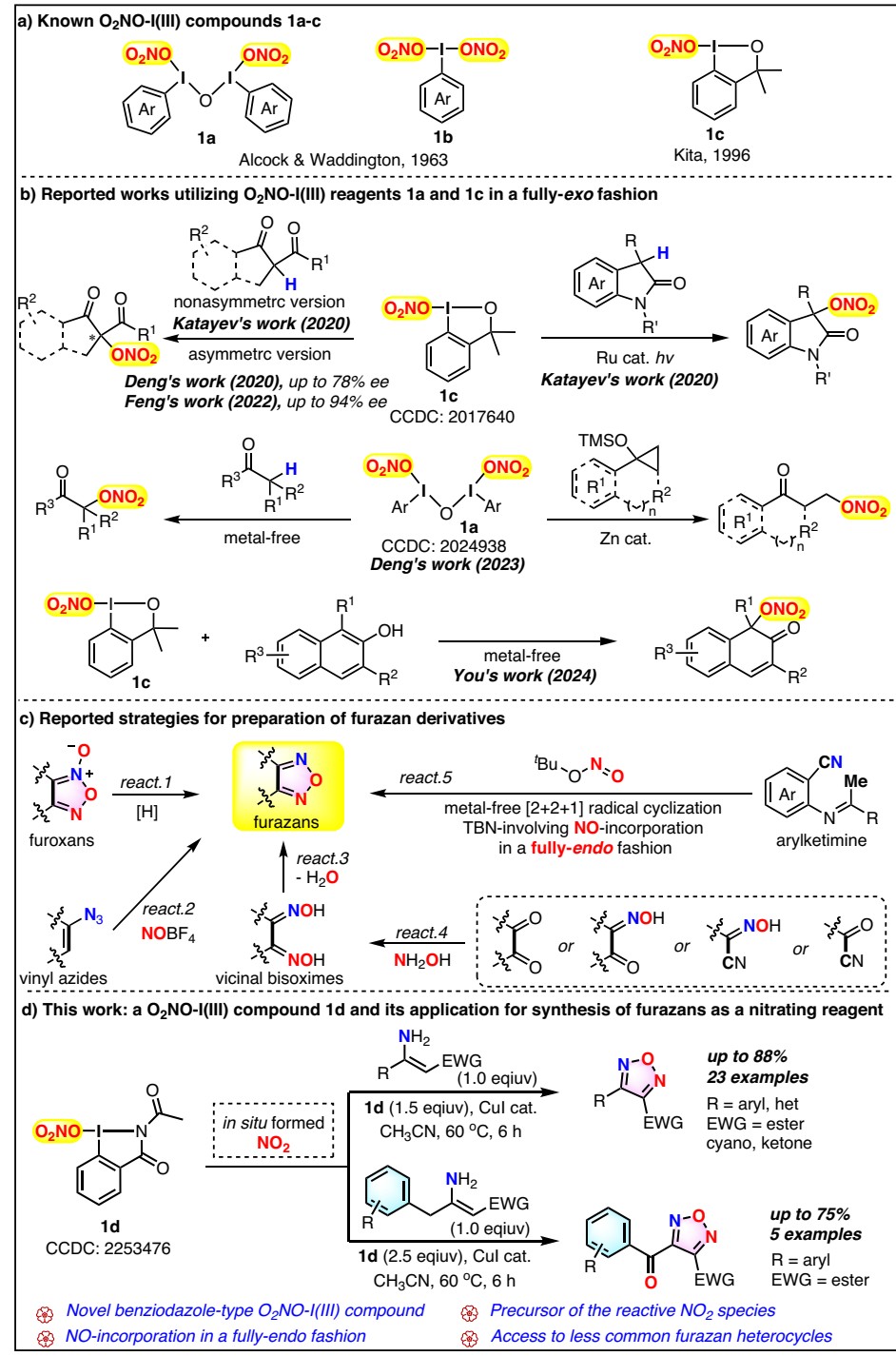

realizing the synthesis of a series of furazan-fused quinolines by employing *tert*-butyl nitrite (ᵗBuONO) to incorporate NO moiety into furazan framework in a fully-*endo* pattern (Fig. 1c, react. 5)[65]. Although all the above approaches have their respective merits in obtaining the corresponding furazan derivatives, the development of novel synthetic routes to access this unique heterocycle should still be of important synthetic value.

Here, we reported that benziodazole-type O₂NO-I(III) **1d**, being a hypervalent O₂NO-containing iodine(III) compound, could react with β-monosubstituted enamines to trigger a copper-catalyzed radical nitration/cyclization/dehydration cascade, providing an alternative protocol to access the exclusive furazan heterocycles (Fig. 1d). Differing from the previous nitrooxylation reactions enabled by the existing O₂NO-I(III) reagents **1a** and **1c**, O₂NO-I(III) compound **1d** in this work was used as nitrating reagent to incorporate its NO moiety to furazan skeleton in a fully-*endo* pattern.

## Results and discussion

In order to further enrich the type of hypervalent iodine(III) reagents[66,67], we were interested in investigating the preparation of a benziodazole-bearing O₂NO group. Following the general procedure[66–69], a ligand exchange reaction of benziodazole-type Cl-I(III) compound **1e** with silver nitrate (AgNO₃) was conducted in dried chloroform under an N₂ atmosphere. The reaction afforded the expected benziodazole-type O₂NO-I(III) **1d** feasibly in 93% yield as a light yellow solid, which is stable for several months when stored at 0 °C in the absence of light (Fig. 2). Thermogravimetry-differential thermal analysis (TG-DTS) showed that compound **1d** decomposed at

**Fig. 2 | Preparation of O$_2$NO-containing benzio-dazole-type I(III) 1d.** Conversion to reagent **1d** from **1e** via ligand exchange reaction and single-crystal X-ray structure of **1d**.

180.7 °C (for details see Supplementary Data 3). Furthermore, a single crystal of **1d** was grown in a mixed solvent of chloroform/*n*-hexane at room temperature, and it crystallized in the monoclinic space group $P2_1/c$ with $Z = 4$. An X-ray crystal analysis of compound **1d** (Fig. 2) revealed a distorted T-shape geometry like the common hypervalent $\lambda^3$-iodanes with an O11–I10–N16 bond angle of 162.32(8)° and I–ONO$_2$ bond length of 2.336(2) Å. The length of the observed I–ONO$_2$ bond in compound **1d** is longer than its analogous 1a and 1c, i.e., 2.311(3) Å[30], and 2.283(2) Å[27], respectively, suggesting reduced covalent character. Compound **1d** also possesses a planar geometry, as indicated by the torsion angles O14–N13–O11–I10 (8.7(3)°), I10–C19–C18–C20 (2.0(3)°), and O17–C20–N16–C21 (8.0(5)°) (for details see Supplementary Data 4).

Initially, our efforts were focused on studying the feasibility of the nitrooxylation reaction of O$_2$NO-I(III) **1d** with β-monosubstituted

**Table 1 | Optimization of the reaction conditions$^{a,b}$**

| Entry | 1d (x eqiuv) | Catalyst | Solvent (mL) | T (°C) | Yield (%)$^b$ |
|---|---|---|---|---|---|
| 1 | 1.5 | CuI | MeCN | 50 | 72 |
| 2 | 1.5 | CuI | DCE | 50 | 43 |
| 3 | 1.5 | CuI | 1,4-dioxane | 50 | 59 |
| 4 | 1.5 | CuI | THF | 50 | nd |
| 5 | 1.5 | CuI | DMF | 50 | nd |
| 6 | 1.5 | CuI | HFIP | 50 | nd |
| 7 | 1.5 | CuBr | MeCN | 50 | 66 |
| 8 | 1.5 | CuSCN | MeCN | 50 | 63 |
| 9 | 1.5 | CuCl | MeCN | 50 | 53 |
| 10 | 1.5 | Cu$_2$O | MeCN | 50 | 58 |
| 11 | 1.5 | CuBr$_2$ | MeCN | 50 | 60 |
| 12 | 1.5 | Cu(OAc)$_2$ | MeCN | 50 | 31 |
| 13 | 1.5 | Cu(OTf)$_2$ | MeCN | 50 | 47 |
| 14 | 1.5 | FeBr$_2$ | MeCN | 50 | 67 |
| 15 | 1.5 | PdCl$_2$ | MeCN | 50 | 23 |
| 16 | 1.5 | Mn(OAc)$_2$ | MeCN | 50 | 29 |
| 17 | 1.5 | Ni(acac)$_2$ | MeCN | 50 | 52 |
| 18 | 1.5 | Co(acac)$_2$ | MeCN | 50 | nd |
| 19 | 1.5 | RhCl(PPh$_3$)$_3$ | MeCN | 50 | 14 |
| 20 | 1.5 | none | MeCN | 50 | nd |
| 21 | 1.5 | CuI | MeCN | rt | trace |
| 22 | 1.5 | CuI | MeCN | 30 | 22 |
| 23 | 1.5 | CuI | MeCN | 60 | 80 |
| 24 | 1.0 | CuI | MeCN | 60 | 64 |

$^a$Reaction conditions: **2a** (0.3 mmol, 1.0 equiv), O$_2$NO-I(III) compounds **1d** (x equiv), catalyst (10 mol%), solvent (4 mL), nitrogen atmosphere, T °C. $^b$Isolated yield of **3a**. *nd* no detection.

enamine **2a** in the presence of 10 mol% CuI in acetonitrile at 50 °C under nitrogen atmosphere. Unexpectedly, it was not the nitrooxylating product but the heterocyclic furazan **3a** that was produced and isolated in 72% yield (Table 1, entry 1). The results of a solvent screening revealed that the reaction in other solvents, including DCE and 1.4-dioxane led to inferior yields of **3a** (Table 1, entries 2–3), while no desired product was observed when THF, DMF, or HFIP was used (Table 1, entries 4–6). The following catalysts screening showed that the reaction proceeded with significant efficiency when CuBr or CuSCN was applied (Table 1, entries 7–8). However, when other copper reagents including CuCl, Cu$_2$O, CuBr$_2$, Cu(OAc)$_2$, or Cu(OTf)$_2$ were used, product **2a** was obtained in relatively lower yield in each case (Table 1, entries 9–13). Other metal additives including FeBr$_2$, PdCl$_2$, Mn(OAc)$_2$, Ni(acac)$_2$, Co(acac)$_2$, and RhCl(PPh$_3$)$_3$ were also investigated. All of them were proved to be compatible with this reaction except Co(acac)$_2$ (for details see Supplementary Table S1). The result of a control reaction conducted in the absence of a copper catalyst provided no desired product, indicating that the copper catalyst is indispensable for the reaction to occur (Table 1, entry 14). Temperature was proved to be another important factor for an efficient transformation, with reaction run at 60 °C afforded the best outcome (Table 1, entries 15–17). Furthermore, the screening on dosage of O$_2$NO-I(III) **1d** indicated that 1.5 equivalents of the hypervalent iodine(III) reagent were necessary for complete consumption of the starting enamine **2a** (Table 1, entries 18–19).

With the optimized reaction conditions in hand (Table 1, entry 17), the substrate scope of this newly established method was evaluated (Fig. 3). A series of substituted enamines **2** were proved to be compatible with the protocol, with all of the reactions proceeded successfully to afford the corresponding substituted furazans **3a–w**. As can be seen from Fig. 3, aryl enamines substituted with electron-neutral, -donating, and -withdrawing groups reacted favorably to afford furazans **3a–g** in moderate to good yields. In addition, halogen-containing substrates were also conveniently converted to the desired products **3h–k** in satisfactory yields. In addition, enamines equipping heterocyclic furyl and thienyl groups, or an aromatic naphthyl substituent, were also suitable for this transformation, yields. Notably, the structure of product **3n** unambiguously provides the corresponding furazans **3l–n** in acceptable to good confirmed by X-ray single-crystal diffraction analysis (for details see Supplementary Data 5). Furthermore, the reaction of substrates bearing other alkoxycarbonyl substituents, such as butoxycarbonyl and ethoxycarbonyl group, also proceeded well with good efficiency (**3o–r**). Strikingly, the method could also be well applied to enamines containing the analogous electron-withdrawing cyano or aroyl substituents (**3s–w**). The utility of this method was further demonstrated by the gram-scale synthesis of compound **3a** in a yield of 67% when 10 mmol of **2a** was used under optimized conditions.

To our surprise, when the reaction of substrate **4a** with a menaphthyl moiety was conducted under the above-optimized reaction conditions, it was not the expected menaphthyl furazan but the benzylic CH$_2$-oxidized compounds, i.e., naphthoyl furazan **5a** as well as its precursor **5a'** that were isolated in a yield of 23% and 55%, respectively (Fig. 4, entry 1). Further study revealed that reaction of menaphthyl-substituted enamine **4a** with 1.0 equiv of O$_2$NO-I(III) **1d** in the presence of CuI catalyst under nitrogen atmosphere gave 89% naphthoyl-containing enamine **5a'** (Fig. 4, entry 2), which could be further converted to product **5a** under standard reaction conditions (see SI for details). Considering above facts as well as the result of the control reaction (see SI for details) where trace yield of **5a'** was formed in the absence of CuI catalyst (with most of starting materiel **4b** unconsumed),

**Fig. 3 | Substrate scope study for synthesis of furazans 3.** [a] Reaction conditions: enamine **2** (0.3 mmol, 1.0 equiv), O₂NO-I(III) **1d** (0.45 mmol, 1.5 equiv), CuI (10% mol) in acetonitrile (4 mL) under N₂ atmosphere at 60 °C for 6 h. [b] Isolated yield. [c] Gram-scale synthesis of **3a**, 67% (10 mmol of **2a** was used).

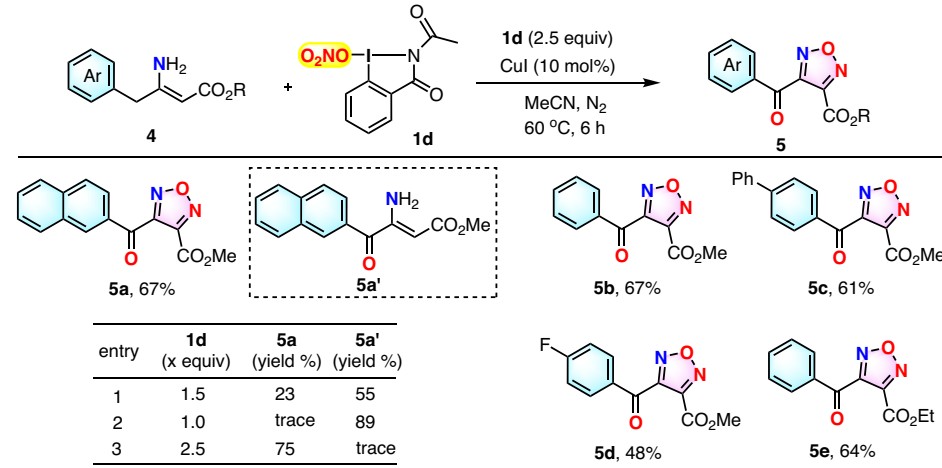

**Fig. 4 | Substrate scope study for synthesis of furazans 5.** [a] Reaction conditions: enamine **4** (0.3 mmol, 1.0 equiv), O₂NO-I(III) **1d** (0.75 mmol, 2.5 equiv), CuI (10% mol) in acetonitrile (4 mL) under N₂ atmosphere at 60 °C for 6 h. [b] Isolated yield.

| entry | 1d (x equiv) | 5a (yield %) | 5a' (yield %) |
|-------|--------------|--------------|---------------|
| 1 | 1.5 | 23 | 55 |
| 2 | 1.0 | trace | 89 |
| 3 | 2.5 | 75 | trace |

**Fig. 5 | Further derivatization of the obtained furazans.** Conversion to furazans 6–7 from compounds **3b** and **3k** via amidation and substitution, respectively.

**Fig. 6 | Mechanism investigation. a** Generation of co-product S1 under standard conditions. **b** Radical-trapping experiments. **c** Investigation on reactive $NO_2$ generated in situ.

we tentatively presumed that enamine **4a** was first oxidized to benzoyl enamine **5a'** by $O_2NO$-I(III) **1d** assisted by CuI catalyst and then the formed **5a'** was converted to product **5a** by further reacting with $O_2NO$-I(III) **1d**. Thus, a larger amount of **1d** was employed to facilitate the complete conversion of enamine **4a** to furazan **5a**. When the amount of $O_2NO$-I(III) **1d** was increased to 2.5 equivalents, a 75% yield of furazan **5a** was attained (Fig. 4, entry 3). Under the most optimal conditions, other benzyl enamines were investigated, and they were all converted to the corresponding furazans **5b–e** with acceptable yields (Fig. 4).

Derivatization of the obtained furazan derivatives was carried out to prove the utility of this method (Fig. 5). To our delight, furazan **3b** could be further transformed into compound **6** via the one-pot two-step amidation[70]. In addition, azide **7** could be achieved from furazan **3k** through nucleophilic substitution reaction[71,72]. Both of the two transformations provided access to new derivatized furazan-containing molecules, demonstrating the stability of the exclusive furazan skeleton under the respective reaction conditions.

To understand the mechanism of this copper-catalyzed $O_2NO$-I(III) **1d**-mediated transformation, a series of control experiments were conducted (Fig. 6). First, the reaction of substrate **2a** under standard conditions produced co-product $N$-acetyl-2-iodobenzamide **S1** in high yield (based on starting material **2a**) (Fig. 6a). Then radical scavenger was introduced to investigate whether the reaction adopts a radical pathway. Specifically, when 1.5 equivalents of TEMPO was employed under standard conditions, the transformation was almost completely inhibited (Fig. 6b). Next, a radical clock experiment was carried out by introducing 1.5 equivalents of compound **8** to the reaction of substrate **2a** under standard reactions, and it was found that furazan **3a**, nitrated compounds **9** and **10** were isolated in yield of 26%, 44%, 17%, respectively (Fig. 6b). The outcomes of the above experiments strongly indicate that reaction process might encompass a radical pathway, and the reactive $NO_2$ species[73–76] might be a crucial intermediate formed in situ from $O_2NO$-I(III) **1d** during the process. To corroborate whether $NO_2$ was the intermediate, control experiment by replacing $O_2NO$-

I(III) **1d** with exogenous brown $NO_2$ gas, generated from the known reaction[77,78] of copper powder and concentrated nitric acid, were conducted (Fig. 6c). To our delight, furazan **3a** was obtained in 88% from the reaction of treating substrate **2a** with $NO_2$ gas in presence of $CuBr_2$ catalyst in acetonitrile at 60 °C for 0.5 h, while no **3a** was detected when no copper catalyst was used. The result of the above control experiment strongly supports our assumption that $NO_2$ is the reactive species that enables the nitration reaction to occur.

Based on the above results as well as the previous reports[73–76,79–83], a plausible radical pathway including two parts (the formation of $NO_2$ and the following $NO_2$-radical addition/cyclization/dehydration cascade) was proposed for this transformation (Fig. 7). Initially, homolysis[79–83] of $O_2NO$-I(III) **1d** under heating gives $O_2NO$ radical and $N$-radical **A1**. Single electron oxidation of CuI by **A1** affords Cu(II) species **A2**. Meanwhile, dimerization of the generated $O_2NO$ radical generates intermediate **B**, which is unstable and undergoes dissociation to release oxygen gas as well as $NO_2$ molecule, a reactive radical species that can dimerize into $N_2O_4$. Then, the radical addition of $NO_2$ to the C–C double bond of enamine **2a** furnishes radical species **C**. Next, one H radical of intermediate **C** is captured by Cu(II) species **A2**, leading to the formation of imine **D** as well as Cu(III) species **A3**, which undergoes reductive elimination to form $N$-acetyl-2-iodobenzamide **S1** and CuI.

Next, two pathways (Fig. 7, the path a and b) were postulated for the formation of furazan **3a** from intermediate **D**. In path a, enamine **E** was formed from imine **D** via tautomerism first. Then nucleophilic attack of the nitrogen atom of enamino moiety in intermediate **E** to its oxygen center of nitrone gave the cyclized intermediate **F**. Subsequent tautomerization of **F** achieved via the system of **S1/S2** and following dehydration of the resulting intermediate **H** gave product **3a**. While in path b, the intramolecular attack of oxygen atom of nitrone in intermediate **D** to nitrogen center of its imino moiety, with the concomitant formation of C–C double bond and cleavage of C–N bond occurred first to give intermediate **I**. Then intramolecular cyclization of **I** provided the cyclized intermediate **J**, which underwent

**Fig. 7 | Plausible mechanism.** The formation of NO$_2$ and the following NO$_2$-radical addition/cyclization/dehydration cascade.

**Fig. 8 | Reaction utilizing enaminone 11 as starting material.** Compound **11** was also a suitable substrate for our nitration/cyclization/dehydration cascade reaction.

similar tautomerization and following dehydration of the resulting intermediate **H** to afford furazan **3a**.

Finally, enaminone **11** was also examined to explore whether it is applicable to this radical nitration/cyclization/dehydration cascade reaction. The results showed that enamine **11** was equally applicable for this transformation under standard conditions, furnishing furazan **5e** in 76% yield (Fig. 8). Interestingly, the two N atoms in product **5e** originate from the amino moiety of enamine **11** and the nitroxyl moiety of O$_2$NO-I(III) **1d**, respectively, but in a completely reversed pattern to the ones of **5e** generated from **4e** (Fig. 4).

In conclusion, we prepared a benziodazole-type hypervalent O$_2$NO-I(III) compound **1d** and had it applied to the synthesis of a series of exclusively heterocyclic furazans from β-monosubstituted enamines via an unprecedented copper-catalyzed radical nitration/cyclization/dehydration cascade. Differing from the existing O$_2$NO-I(III) reagents that have been uniformly used as nitrooxylating reagents for introducing O$_2$NO moiety, the O$_2$NO-I(III) **1d** described in this work can be regarded as a nitrating reagent and incorporate its NO moiety to furazan skeleton in a fully-*endo* pattern. Furthermore, the current method also provides an alternative approach, which is in nature different from the existing strategies[34,49–65], to the biologically interesting furazan heterocycles.

## Methods

### Procedure for synthesis of O$_2$NO-Iodine(III) 1d

To a 200 mL two-necked round-bottomed flask were added compound **1e** (3.23 g, 10 mmol, 1.0 equiv), AgNO$_3$ (3.4 g, 20 mmol, 2.0 equiv) and dried CHCl$_3$ (70 mL) under N$_2$ atmosphere. The reaction mixture was stirred at room temperature in the dark for 3.5 days. The mixture was then filtered through a pad of Celite and washed with CHCl$_3$ (1000 mL). The solvents were concentrated in a vacuum to give compound **1d** as a white solid.

### General procedure for the synthesis of substituted furazans 3 and 5

To a 20 mL Schlenk tube equipped with a stirrer was added β-monosubstituted enamine **2** (0.3 mmol, 1.0 equiv), O$_2$NO-I(III) **1d** (0.45 mmol, 1.5 equiv) and CuI (0.03 mmol, 6 mg, 10 mol%) under N$_2$ atmosphere, followed by addition of acetonitrile (4 mL). The tube was screw-capped and stirred at 60 °C. After stirring for 6 h, the reaction mixture was diluted with dichloromethane, filtered through a pad of Celite, and concentrated in a vacuum. The residue was purified with silica gel chromatography (PE/EtOAc) to afford furazans **3**. (When enamine **4** and O$_2$NO-I(III)

**1d** (0.75 mmol, 2.5 equiv) were employed as starting materials under the above conditions, substituted furazans **5** was obtained).

## Data availability

All data generated during this study are included in this article and Supplementary Information. Experimental procedure, conditions optimization and product characterization are provided in the Supplementary Information. The NMR spectra of all compounds are available in Supplementary Data 1. The infrared spectra of compound **1d** is available in Supplementary Data 2. The TG-DTA and DSC analysis of compound **1d** is available in Supplementary Data 3. The crystallographic data for compounds **1d**, **3n**, and **10** can be obtained free of charge from the Cambridge Crystallographic Data Centre (CCDC) under deposition numbers 2253476 (**1d**, Supplementary Data 4), 2255860 (**3n**, Supplementary Data 5) and 2320692 (**10**, Supplementary Data 6), respectively. These data can be obtained free of charge from the Cambridge Crystallographic Data Centre via www.ccdc.cam.ac.uk/data_request/cif.

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

## Acknowledgements
YF.D. acknowledges the National Natural Science Foundation of China (No. 22071175) and X.L thanks Tianjin Research Innovation Project for Postgraduate Students (No. 2021YJSB196). We also thank Yan Gao and Xiangyang Zhang [AIC, SPST/TJU] for providing the analysis support.

## Author contributions
Z. F. Y. and Y. F. D. conceived and designed the experiments; Z. F. Y. carried out most of the experiments; J. X. carried out the single-crystal X-ray experiments; Y. L. S., X. M. L., and B. H. J. analyzed data; Z.F.Y. and Y.F.D. directed the project and wrote the paper.

## Competing interests
The authors declare no competing interests.
