## [Peer Review File · Communications Chemistry]

This manuscript has been previously reviewed at another Nature Portfolio journal. This document only contains reviewer comments and rebuttal letters for versions considered at *Communications Chemistry*.Reviewers' comments:

Reviewer #1 (Remarks to the Author):

I was asked to evaluate the computational aspects of this manuscript for plausibility of the computed mechanism of the reaction. Its pretty simple. Its wrong. While the authors have used OK computational methods, the energies they have calculated are wildly inconsistent with the reported experimental conditions (60 deg C, 6 h). For both Path A and B the calculated barriers are far too high for these conditions. If one imagines that the authors have reported kj values and messed up and thought it was kcal (its very clearly stated in multiple cases its kcal so I doubt this is the case) than it might be more consistent. However, if the authors confirm the calculated values are indeed in kcal/mol the pathway as calculated is implausible. The calculated barrier is at least two times as high as might be something that approaches plausible under those experimental conditions. The work needs to be redone or omitted from the manuscript. "We don't know the mechanism" is not a paper killer.

Reviewer #2 (Remarks to the Author):

Recommendation: Suitable for Communications Chemistry after minor revisions.

Du and co-workers describe the synthesis of a new I(III)-ONO₂ hypervalent iodine reagent used as nitrating reagent for the synthesis of furazans under copper catalysis. Conceptually, the use of hypervalent iodine reagents allows the safe introduction of a nitro group avoiding the use of harsh conditions such as concentrated nitric acid. When using this new reagent with β -monosubstituted enamines, furazans were obtained. The authors also investigated the plausible mechanism of the reaction, supported by control experiments and DFT calculations.

A straightforward synthesis of I(III)-ONO₂ is presented by the authors in excellent yield. An X-ray analysis is also presented for this new reagent. It was then applied for the synthesis of furazans under copper-catalyzed conditions. Various enamines could be transformed in the corresponding furazans. The reaction could be extended to enaminones. The products were obtained in good yields. The protocol described here uses simple CuI and safe new I(III)-ONO₂ avoiding the use of nitric acid.

Therefore, the development and application of this new I(III)-ONO₂ is a very well-done work and nicely presented. It is also worthy to note the quality of the SI. Consequently, I am recommending the publication in Communications Chemistry after the minor revisions listed below.

Revision needed:

- Stability

The authors show the stability of the newly made hypervalent iodine reagent, however, the qualification of bench stable ("bench-stable for several months when stored at 0 °C in the absence of light") is not adequate here as it requires to be stored in a fridge in the dark. This should be modified to "stable for several months when stored at 0 °C in the absence of light".

- Length bond analysis

"The length of the observed I-ONO₂ bond is significantly longer than its average I-O bond length of 2.14[34] and longer than the I-ONO₂ bond length in the analogous O₂NO containing I(III) 1a and 1c, i.e., 2.311(3) Å and 2.283(2) Å respectively, suggesting reduced covalent character." In this comparison, it should appear clearly that the new compound is compared to a different class of HIR, PhI(OR)₂ which are not cyclic.

- Catalyst type

"Other metal additives including FeBr₂, PdCl₂, Mn(OAc)₂, Ni(acac)₂, Co(acac)₂ and RhCl(PPh₃)₃ were also investigated, but none was proved to be compatible with this reaction (see SI for details)."

As shown in the SI, only Co(acac)₂ is not affording product. All the other catalysts are giving product, in very good yields for some of them.

- Equivalent of 1d

It is worth trying 1.25 equivalents of nitrating HIR 1d in the optimization.

- Benzylic oxidation

"To our surprise, when the reaction of substrate 4a with a menaphthyl moiety was conducted under above optimized reaction conditions, it was not the expected menaphthyl furazan but naphthoyl furazan 5a". Here, the main point is not the presence of the naphthyl, but the presence

of an oxidizable benzylic CH₂ position. This should be emphasized.

- Radical scavenging with TEMPO

When using TEMPO as radical scavenger, was TEMPO-adduct observed?

- Mechanistic studies

The mechanistic studies and proposal are solid. However, Scheme 4 is very crowded and it would be useful to simplify it. Indeed, from D to F, as it is a classic enamine-imine tautomerism, it is probably not necessary to draw the two intermediates in between. Same from G to I as it is just a prototropy. Same from I to N. Also, the role of S1/S2 is already well highlighted in the DFT scheme.

In absence of copper, was intermediate C or D observed?

In the DFT scheme, there is a mistake in structure M with the double bond between N=O compared to Scheme 4.

- Typos

-“Compound 1d also possesses a planar geometry, as indicated by the...” ◊ 1d should be in bold

- Table 1, entry 12: CuI(OAc)₂ ◊ Should be Cu(OAc)₂

-“To our delight, furazan 3b could be further transformed into compound 7 via the one-pot two-step amidation. In addition, azide 6 could be ...” 6 and 7 are inverted compared to the scheme.

Supplementary Information

The quality of is SI is very high. It is very pleasant to read. Couple of points:

- Retention factors

The retention factors of all compounds purified by flash chromatography should be added with the corresponding eluent used for the TLC.

- I(III)-ONO₂

For the characterization of 1d, a HRMS analysis would be appreciated.

- Starting materials characterization

Although most of the starting enamines are described in the literature, it is always interesting and important to have the characterization included in the article’s SI if they have been made by hand. I recommend adding the proton report for each used starting material used (13C would be a must).

- Reaction time

For several procedures, it is indicated that reaction was stopped after TLC analysis. The exact time of reaction should be added. This is valid for all starting materials, compound 6, procedure with conc. HNO₃.

- Control experiments

For the radical trapping with TEMPO: “The reaction was monitored by TLC, and trace of 3a was observed.” A yield is necessary and not an approximation by TLC. Also, it is indicated that traces were observed, but on the scheme, it is indicated not detected. This needs to be addressed.

- Procedures

For all procedures, the quantity in mg or g should be added next to the mmol and equivalents.

- Schemes

A general comment concerning the structures. I would be easier to read if the molecules were redrawn. For example, from 12 to S1 page S25. Adding S1 structure is recommended

- Spectra

For all NMR spectra, please indicate on the file/picture the nucleus, the solvent and the frequency.
ex: 13C NMR (400 MHz, CDCl₃) of compound 3f

- NMR Spectra provided are in the majority clean. Compounds 3u, 5a and 5d should be purified to obtain cleaner NMR. (1H, 19F and 13C)

- Some NMR provided need to be shimmed: ex: 6

Reviewer #3 (Remarks to the Author):

I co-reviewed this manuscript with one of the reviewers who provided the listed reports. This is part of the Communications Chemistry initiative to facilitate training in peer review and to provide appropriate recognition for Early Career Researchers who co-review manuscripts.

Reviewer #4 (Remarks to the Author):

Du and co-workers report the formation of furazans using a iodine(III) reagent to transfer a NO moiety to activated enamines. The reaction is interesting and the experimental part, including control experiments and a reasonable scope, is clearly reported. I am positively surprised by the mechanistic experiments (especially the NO₂ gas experiment), but the computational part is fundamentally wrong and should be recalculated (or removed if the editor thinks that the experimental part is sufficient for publication). Therefore, I would recommend rejection of the manuscript in its current state and eventual resubmission after addressing the following major concern:

- The authors performed a series of DFT calculations starting from species D. The addition of NO₂ to the reactant has not been studied and the reduction process is presumed but not demonstrated. Then, the computed free energy profile shows a series of activation free energy barriers above 40 kcal/mol, which are clearly not reasonable for a thermal process and cannot explain the reactivity at all. In fact, the iodine species, resulting after the generation of NO₂, is used as acid-base proton shuttle, which is not consistent with the experimental observations. According to the reported experiments, Cu salt is crucial for both the iodine mediated experiment and the NO₂ gas experiment (Scheme 3C). This strongly points out that the iodine(III) role is to release NO₂ gas in the reaction media and then, NO₂ + Cu(II) salt combination is the actual responsible of the reactivity. Therefore, Cu(II) species should be explicitly included in the calculations. Also, the role of Cu(II)-Cu(III) as mediator in single electron transfer steps or assistance in the N-O bond breaking step is known and should be included in the calculations. I do recommend the authors to start the calculations from the initial enamine substrate + NO₂ + Cu(II) salt (ignoring the role of iodine(III), which is clearly demonstrated by the experimental work) if they really want to figure out the mechanism of the process.

Reviewer's suggestions

Reviewer 1's comments:

I was asked to evaluate the computational aspects of this manuscript for plausibility of the computed mechanism of the reaction. It's pretty simple. It's wrong. While the authors have used OK computational methods, the energies they have calculated are wildly inconsistent with the reported experimental conditions (60 deg C, 6 h). For both Path A and B the calculated barriers are far too high for these conditions. If one imagines that the authors have reported kj values and messed up and thought it was kcal (it's very clearly stated in multiple cases its kcal so I doubt this is the case) than it might be more consistent. However, if the authors confirm the calculated values are indeed in kcal/mol the pathway as calculated is implausible. The calculated barrier is at least two times as high as might be something that approaches plausible under those experimental conditions. The work needs to be redone or omitted from the manuscript. "We don't know the mechanism is not a paper killer.

Response: We thank this Reviewer for putting forward this valuable suggestion. According to the suggestions, the DFT calculations for both Path a and b were omitted from the revised manuscript.

Reviewer 2's comments:

Recommendation: Suitable for Communications Chemistry after minor revisions.

Du and co-workers describe the synthesis of a new I(III)-ONO₂ hypervalent iodine reagent used as nitrating reagent for the synthesis of furazans under copper catalysis. Conceptually, the use of hypervalent iodine reagents allows the safe introduction of a nitro group avoiding the use of harsh conditions such as concentrated nitric acid. When using this new reagent with β-monosubstituted enamines, furazans were obtained. The authors also investigated the plausible mechanism of the reaction, supported by control experiments and DFT calculations.

A straightforward synthesis of I(III)-ONO₂ is presented by the authors in excellent yield. An X-ray analysis is also presented for this new reagent. It was then applied for the synthesis of furazans under copper-catalyzed conditions. Various enamines could be transformed in the corresponding furazans. The reaction could be extended to enaminones. The products were obtained in good yields. The protocol described here uses simple CuI and safe new I(III)-ONO₂ avoiding the use of nitric acid.

Therefore, the development and application of this new I(III)-ONO₂ is a very well-done work and nicely presented. It is also worthy to note the quality of the SI. Consequently, I am

recommending the publication in Communications Chemistry after the minor revisions listed below.

Revision needed:

1) Stability

The authors show the stability of the newly made hypervalent iodine reagent, however, the qualification of bench stable (“bench-stable for several months when stored at 0 °C in the absence of light”) is not adequate here as it requires to be stored in a fridge in the dark. This should be modified to “stable for several months when stored at 0 °C in the absence of light”.

Response: We thank this Reviewer for putting forward this valuable comment. According to the suggestions, the above issue has been corrected as follows:

“The reaction afforded the expected benziodazole-type O₂NO-I(III) **1d** feasibly in 93% yield as a light yellow solid, which is **stable** for several months when stored at 0 °C in the absence of light (Scheme 1)”

2) Length bond analysis

“The length of the observed I–ONO₂ bond is significantly longer than its average I–O bond length of 2.14^[34] and longer than the I–ONO₂ bond length in the analogous ONO₂ containing I(III) **1a** and **1c**, i.e., 2.311(3) Å and 2.283(2) Å respectively, suggesting reduced covalent character.” In this comparison, it should appear clearly that the new compound is compared to a different class of HIR, PhI(OR)₂ which are not cyclic.

Response: We thank the Editor for putting forward this valuable comment, which is a good thinking perspective that we have neglected. Actually, what we originally wanted to emphasize is the change of the observed I–ONO₂ bond length compared to the existing I–ONO₂ containing HIRs. For clarity purpose, according to the suggestions, the above issue has been corrected as follows:

“The length of the observed I–ONO₂ bond **in compound 1d is significantly** longer than its **average I–O bond length of 2.14^[34] and longer than the I–ONO₂ bond length in the** analogous **O₂NO-containing I(III)–1a and 1c**, i.e., 2.311(3) Å,^[16] and 2.283(2) Å^[13] respectively, suggesting reduced covalent character.”

3) Catalyst type

“Other metal additives including FeBr₂, PdCl₂, Mn(OAc)₂, Ni(acac)₂, Co(acac)₂ and RhCl(PPh₃)₃ were also investigated, but none was proved to be compatible with this reaction (see SI for details).”

As shown in the SI, only Co(acac)₂ is not affording product. All the other catalysts are giving product, in very good yields for some of them.

Response: We thank this Reviewer for putting forward this valuable comment and we are terribly sorry for this typo. According to the suggestions, the above description has been corrected as follows:

“Other metal additives including FeBr₂, PdCl₂, Mn(OAc)₂, Ni(acac)₂, Co(acac)₂ and RhCl(PPh₃)₃ were also investigated. All of them were proved to be compatible with this reaction except Co(acac)₂ (see SI for details).”

4) Equivalent of **1d**

It is worth trying 1.25 equivalents of nitrating HIR **1d** in the optimization.

Response: We thank this Reviewer for putting forward this valuable comment. According to the suggestions, we added the reaction trying 1.25 equivalents of nitrating HIR **1d** in the optimization Table 1, as entry 18.

Table 1. Optimization of the reaction conditions.^[a]

Entry	1d (x equiv)	Catalyst	Solvent	T (°C)	Yield (%) ^[b]
1	1.5	CuI	MeCN	50	72
2	1.5	CuI	DCE	50	43
3	1.5	CuI	1,4-dioxane	50	59
4	1.5	CuI	THF	50	nd
5	1.5	CuI	DMF	50	nd
6	1.5	CuI	HFIP	50	nd
7	1.5	CuBr	MeCN	50	66
8	1.5	CuSCN	MeCN	50	63
9	1.5	CuCl	MeCN	50	53
10	1.5	Cu ₂ O	MeCN	50	58
11	1.5	CuBr ₂	MeCN	50	60
12	1.5	Cu(OAc) ₂	MeCN	50	31
13	1.5	Cu(OTf) ₂	MeCN	50	47
14	1.5	none	MeCN	50	nd
15	1.5	CuI	MeCN	rt	trace

16	1.5	CuI	MeCN	30	22
17	1.5	CuI	MeCN	60	80
18	1.25	CuI	MeCN	60	73
19	1.0	CuI	MeCN	60	64

^[a] Reaction conditions: **2a** (0.3 mmol, 1.0 equiv), O₂NO-I(III) **1d** (x equiv), Cu catalyst (10 mol%), solvent (4 mL), N₂ atmosphere, T °C, 6 h. ^[b] Isolated yield of **3a**. nd = not detected.

For above added entry, corresponding description have been corrected as follows:

“Furthermore, the screening on dosage of O₂NO-I(III) **1d** indicated that 1.5 equivalents of the hypervalent iodine(III) reagent was necessary for a complete consumption of the starting enamine **2a** (Table 1, entries 18-19).”

5) Benzylic oxidation

“To our surprise, when the reaction of substrate **4a** with a benzylic CH₂ position was conducted under above optimized reaction conditions, it was not the expected menaphthyl furazan but naphthoyl furazan **5a**”. Here, the main point is not the presence of the naphthyl, but the presence of an oxidizable benzylic CH₂ position. This should be emphasized.

Response: We thank this Reviewer for having this question pointed out. According to the suggestions, the above issues have been corrected as follows:

“To our surprise, when the reaction of substrate **4a** with a menaphthyl moiety was conducted under above optimized reaction conditions, it was not the expected menaphthyl furazan but the benzylic CH₂-oxidized compound, i.e., naphthoyl furazan **5a** as well as its precursor **5a'** that were isolated in a yield of 23% and 55%, respectively (Table 3, entry 1).”

6) Radical scavenging with TEMPO

When using TEMPO as radical scavenger, was TEMPO-adduct observed?

Response: We thank the Reviewer for this valuable suggestion. However, we are sorry to see that TEMPO-adduct was not observed when TEMPO was used as radical scavenger. However, this could not rule out the possibility of a radical process as a radical clock experiment with 2-(1-cyclopropylvinyl)naphthalene **8** did work. The corresponding results were shown in Scheme 3B of the original manuscript.

7) Mechanistic studies

The mechanistic studies and proposal are solid. However, Scheme 4 is very crowded and it would be useful to simplify it. Indeed, from **D** to **F**, as it is a classic enamine-imine tautomerism, it is probably not necessary to draw the two intermediates in between. Same from **G** to **I** as it is just a prototropy. Same from **L** to **N**. Also, the role of **S1/S2** is already well highlighted in the DFT scheme.

In absence of copper, was intermediate **C** or **D** observed?

In the DFT scheme, there is a mistake in structure **M** with the double bond between N=O compared to Scheme 4.

Response: We thank this Reviewer for having these questions pointed out. We had performed the control reaction in the absence of copper, but no intermediate **C** or **D** was detected. Furthermore, we decide to delete the DFT calculations according the suggestions from other reviewers. According to the suggestions of this reviewer, the above issue have been corrected as follows:

~~“In order to well explain the transformation of imine **D** into furazan **3a**, two pathways (Scheme 4, path a and b) were postulated. Path a encompasses the first conversion of imine **D** to enamine **F**, then the formation of cyclized intermediate **G** from the nucleophilic attack of nitrogen atom of enamino moiety to oxygen center of nitrone in intermediate **F**, and the subsequent tautomerization (via intermediates **H-J**) and dehydration to give product **3a**. While in path b, the intramolecular attack of oxygen atom of nitrone to nitrogen center of imino moiety in intermediate **D**, with the concomitant formation of C-C double bond and cleavage of C-N bond occurs first to give intermediate **K**. Then intramolecular cyclization of **K** provides the cyclized intermediate **L**, which undergoes similar tautomerization (via intermediates **M-J**) and dehydration to afford furazan **3a**.~~

~~To gain more evidence for the most possible mechanistic pathway, we performed density functional theory (DFT) calculations on the reaction of imine **D** under standard conditions (Scheme 5). As can be seen from the calculation results, in path a, **D** first bounds with **S1** to form intermediate **CP**, which undergoes deprotonation to form intermediate **R-E**, a resonance species of **E**, through **TSCP-E** with free energy barrier of 24.3 kcal/mol. Further a protonation occurs at nitrogen atom of **R-E** to produce the cyclized precursor **F**. Then intramolecular cyclization of **F** affords intermediate **G** through the transition state **TSF-G** with a high energy barrier of 87.3 kcal/mol. Subsequently, a series of deprotonation protonation reactions of intermediate **G** and the final dehydration give the final product **3a**. Comparatively in path b, **D** first undergoes cyclization to produce intermediate **L** through transition state **TSD-L**, with a relatively lower free energy barrier of 66.8 kcal/mol. Notably, this step occurs not through the~~

imaged intermediate **K** drawn in Scheme 4. Then intermediate **J** is obtained after several deprotonation-protonation reactions of **L**. Judged by the free energy barrier, path b should be a more favorable mechanistic pathway for the cyclization process.

‘Next, two pathways (Scheme 4, path a and b) were postulated for the formation of furazan **3a** from intermediate **D**. In path a, enamine **E** was formed from imine **D** via tautomerism first. Then nucleophilic attack of nitrogen atom of enamino moiety in intermediate **E** to its oxygen center of nitron gave the cyclized intermediate **F**. Subsequent tautomerization of **F** achieved via the system of S1/S2 and following dehydration of the resulting intermediate **H** gave product **3a**. While in path b, the intramolecular attack of oxygen atom of nitron in intermediate **D** to nitrogen center of its imino moiety, with the concomitant formation of C-C double bond and cleavage of C-N bond occurred first to give intermediate **I**. Then intramolecular cyclization of **I** provided the cyclized intermediate **J**, which underwent similar tautomerization and following dehydration of the resulting intermediate **H** to afford furazan **3a**.’

Scheme 4. Plausible Mechanism.

8) Typos

-“Compound **1d** also possesses a planar geometry, as indicated by the...” **1d** should be in bold

- Table 1, entry 12: CuI(OAc)₂ Should be Cu(OAc)₂

-“To our delight, furazan **3b** could be further transformed into compound **7** via the one-pot two-step amidation. In addition, azide **6** could be ...” **6** and **7** are inverted compared to the scheme.

Response: We thank this Reviewer for having these questions pointed out. According to the suggestions, the typo, i.e., CuI(OAc)₂ (Table 1, entry 12), has been corrected to Cu(OAc)₂. Other typos pointed out by this Reviewer have been corrected as follows:

“Compound **1d** also possesses a planar geometry, as indicated by the torsion angles O14–N13–O11–I10 (8.7(3)°), I10–C19–C18–C20 (2.0(3)°) and O17–C20–N16–C21 (8.0(5)°).”

“To our delight, furazan **3b** could be further transformed into compound **6** via the one-pot two-step amidation.^[34] In addition, azide **7** could be achieved from furazan **3k** by reacting with methyl glycinate through aminolysis of ester.^[35]”

Supplementary Information

The quality of is SI is very high. It is very pleasant to read. Couple of points:

9) Retention factors

The retention factors of all compounds purified by flash chromatography should be added with the corresponding eluent used for the TLC.

Response: We thank this Reviewer for having this question pointed out. According to the suggestions, the retention factors of all compounds purified by flash chromatography have been added with the corresponding eluent used for the TLC. The detail can be seen in the revised Supporting Information.

10) I(III)-ONO₂

For the characterization of **1d**, a HRMS analysis would be appreciated.

Response: We thank this Reviewer for having this question pointed out. Actually, we tried to record HRMS data for compound **1d** but failed. Only the molecular ion peak of the benzodiazole skeleton of compound **1d** was detected by LRMS and HRMS as can be seen in following Figures:

Instrument: Thermo Scientific Q Exactive HF Orbitrap-FTMS

Card Serial Number: E241112

Sample Serial Number: 2020914-yxf-NO3

Operator: Songw

Date: 2024/04/22

Operation Mode: ESI Positive Ion Mode

Elemental composition search on mass 287.9512

m/z= 282.9512-292.9512

m/z	Theo. Mass	Delta (ppm)	RDB equiv.	Composition
287.9512	287.9516	-1.28	6.5	C ₉ H ₇ O ₂ N I
	287.9524	-3.90	1.5	C ₃ H ₇ O ₅ N ₂ ¹⁰ B I

Additionally, similar result was found in the work of Zhang (*Angew. Chem. Int. Ed.*, 2021, 60, 24171, doi.org/10.1002/anie.202108589), who also failed to obtain the HRMS data for their monofluoro-containing hypervalent iodine reagent. Thus the HRMS data of M-ONO₂ has been supplemented in the Supporting Information.

Furthermore, the IR data of compound **1d** has been added in the revised Supporting Information as follows, and we hope this could make up for the deficiency of HRMS analysis of compound **1d**:

“HRMS (ESI): Calcd. for C₉H₇INO₂: 287.9516 (M – ONO₂)⁺; Found: 287.9512. FTIR (neat, cm⁻¹): 3293, 3065, 2925, 2855, 1697, 1649, 1576, 1481, 1442, 1359, 1268, 1149, 991, 866, 784, 729, 653, 586.”

Figure S1. The FTIR spectrum of I(III)-ONO₂ **1d**.

The above HRMS data and FTIR as well as its diagram have been supplemented in the revised Supporting Information.

11) Starting materials characterization

Although most of the starting enamines are described in the literature, it is always interesting and important to have the characterization included in the article's SI if they have been made by hand. I recommend adding the proton report for each used starting material used (¹³C would be a must).

Response: We thank this Reviewer for having this question pointed out. According to the suggestions, we have added the proton and carbon reports and spectra for each used starting material. The detail can be found in the revised Supporting Information.

12) Reaction time

For several procedures, it is indicated that reaction was stopped after TLC analysis. The exact time of reaction should be added. This is valid for all starting materials, compound **6**, procedure with conc. HNO₃.

Response: We thank this Reviewer for putting forward this valuable comment. According to the suggestions, we have checked and added the exact time of every reaction. The detail can be seen in the revised Supporting information.

13) Control experiments

For the radical trapping with TEMPO: “The reaction was monitored by TLC, and trace of **3a** was observed.” A yield is necessary and not an approximation by TLC. Also, it is indicated that traces were observed, but on the scheme, it is indicated not detected. This needs to be addressed.

Response: We thank this Reviewer for having this question pointed out. We are terribly sorry for the confusion caused by our carelessness. According to the suggestions, we have checked and corrected the description about the yield of **3a**. The detail can be seen in the revised Supporting Information.

14) Procedures

For all procedures, the quantity in mg or g should be added next to the mmol and equivalents.

Response: According to the suggestions, we have checked and added the quantity in mg or g next to the mmol and equivalents for all procedures. The detail can be found in the revised Support Information.

15) Schemes

A general comment concerning the structures. I would be easier to read if the molecules were redrawn. For example, from **12** to **S1** page S25. Adding **S1** structure is recommended.

Response: According to the suggestions, we have checked and added all structures in schemes in page 25, and the detail can be seen in the revised Support Information.

16) Spectra

For all NMR spectra, please indicate on the file/picture the nucleus, the solvent and the frequency. ex: ^{13}C NMR (400 MHz, CDCl_3) of compound **3f**

- NMR Spectra provided are in the majority clean. Compounds **3u**, **5a** and **5d** should be purified to obtain cleaner NMR. (^1H , ^{19}F and ^{13}C)

- Some NMR provided need to be shimmed: ex: **6**.

Response: We thank this Reviewer for having this question pointed out. According to the suggestions, we have indicated on the file/picture the nucleus, the solvent and the frequency for all NMR spectra, and also obtained cleaner NMR spectra (^1H , ^{19}F and ^{13}C) of compounds

3u, **5a** and **5d** and provided shimmed NMR of compound **6** again. The details can be found in the revised Supporting Information.

Reviewer 3's comments:

I co-reviewed this manuscript with one of the reviewers who provided the listed reports. This is part of the Communications Chemistry initiative to facilitate training in peer review and to provide appropriate recognition for Early Career Researchers who co-review manuscripts.

Response: We thank the Reviewer for the kind reminder. We have made the extensive revisions to the original manuscript based on the concerns and questions raised by all the Reviewers and the Editorial Office. Please find the details in the revised manuscript and the revised Supporting Information.

Reviewer 4's comments:

Du and co-workers report the formation of furazans using a iodine(III) reagent to transfer a NO moiety to activated enamines. The reaction is interesting and the experimental part, including control experiments and a reasonable scope, is clearly reported. I am positively surprised by the mechanistic experiments (especially the NO₂ gas experiment), but the computational part is fundamentally wrong and should be recalculated (or removed if the editor thinks that the experimental part is sufficient for publication). Therefore, I would recommend rejection of the manuscript in its current state and eventual resubmission after addressing the following major concern:

- The authors performed a series of DFT calculations starting from species **D**. The addition of NO₂ to the reactant has not been studied and the reduction process is presumed but not demonstrated. Then, the computed free energy profile shows a series of activation free energy barriers above 40 kcal/mol, which are clearly not reasonable for a thermal process and cannot explain the reactivity at all. In fact, the iodine species, resulting after the generation of NO₂, is used as acid-base proton shuttle, which is not consistent with the experimental observations. According to the reported experiments, Cu salt is crucial for both the iodine mediated experiment and the NO₂ gas experiment (Scheme 3C). This strongly points out that the iodine(III) role is to release NO₂ gas in the reaction media and then, NO₂ + Cu(II) salt combination is the actual responsible of the reactivity. Therefore, Cu(II) species should be explicitly included in the calculations. Also, the role of Cu(II)-Cu(III) as mediator in single electron transfer steps or assistance in the N-O bond breaking step is known and should be

included in the calculations. I do recommend the authors to start the calculations from the initial enamine substrate + NO₂ + Cu(II) salt (ignoring the role of iodine(III), which is clearly demonstrated by the experimental work) if they really want to figure out the mechanism of the process.

Response: We thank this reviewer for putting forward this valuable comment, which we highly appreciate. According to the suggestions from other Reviewers, we have deleted the DFT calculation parts from our original manuscript. However, we will give full consideration to the valuable suggestion from this Reviewer in our future work.

Accordingly, the original second author, i.e., Chang Xu, who carried out the computation, has also been removed from the author list. All the original authors have seen and approved this adjustment.

Table. A Point-to-point Description of Changes

Location	Original MS	Revised MS
Page 1 In TOC,	Zhifang Yang, ¹ Chang Xu, ² Jun Xu, ¹ Yuli Sun, ¹ Xuemin Li, ¹ Bohan Jia, ¹ Yunfei Du ^{1,*}	Zhifang Yang, Jun Xu, Yuli Sun, Xuemin Li, Bohan Jia, Yunfei Du*
Page 1 In TOC,	¹ Tianjin Key Laboratory for Modern Drug Delivery & High-Efficiency, School of Pharmaceutical Science and Technology, Tianjin University, Tianjin 300072, China. ² Research Center for Chemical Safety & Security and Verification Technology & College of Chemical and Pharmaceutical Engineering, Hebei University of Science and Technology; Shijiazhuang 050018, China. E-mail: duyunfeier@tju.edu.cn	Tianjin Key Laboratory for Modern Drug Delivery & High-Efficiency, School of Pharmaceutical Science and Technology, Tianjin University, Tianjin 300072, China.
Page 1 In TOC,	² School of Pharmaceutical Sciences and Yunnan Key Laboratory of Pharmacology for Natural Products; College of Modern Biomedical Industry, Kunming Medical University, 1168 Western Chunrong Road, Yuhua Street, Chenggong District, Kunming City, Yunnan 650500, China.	delete
Page 1, Abstract	A novel benziodazole-type O ₂ NO-I(III) compound was prepared and its reaction with β-monosubstituted enamines in the presence of CuI could trigger a radical nitration/cyclization/dehydration cascade to provide a series of less explored but biologically interesting furazan heterocycles. Mechanistically, the benziodazole-type O ₂ NO-I(III) compound acts as a nitrating reagent and incorporates its NO moiety into the final furazan product in a fully-endo model, a process of which was proposed to involve nitration, cyclization and dehydration. Control experiments as well as DFT calculations were performed to provide mechanistic insights into this exclusive cascade transformation.	A novel benziodazole-type O ₂ NO-I(III) compound was prepared and its reaction with β-monosubstituted enamines in the presence of CuI could trigger a radical nitration/cyclization/dehydration cascade to provide a series of less explored but biologically interesting furazan heterocycles. Mechanistically, the benziodazole-type O ₂ NO-I(III) compound acts as a nitrating reagent and incorporates its NO moiety into the final furazan product in a fully-endo model, a process of which was proposed to involve nitration, cyclization and dehydration.
Page 2, Scheme 1		Page 2, Paragraph 3	The reaction afforded the expected benziodazole-type O ₂ NO-I(III) 1d feasibly in 93% yield as a light yellow solid, which is bench-stable for several months when stored at 0 °C in the absence of light (Scheme 1). Thermogravimetry-differential thermal	The reaction afforded the expected benziodazole-type O ₂ NO-I(III) 1d feasibly in 93% yield as a light yellow solid, which is stable for several months when stored at 0 °C in the absence of light (Scheme 1). Thermogravimetry-differential thermal analysis (TG-DTS)

	analysis (TG-DTS) showed that compound 1d decomposed at 180.7 °C (see SI for details). Furthermore, a single crystal of 1d was grown in a mixed solvents of chloroform/n-hexane at room temperature, and it crystallized in the monoclinic space group $P2_1/c$ with $Z = 4$. A X-ray crystal analysis of compound 1d (Scheme 1) revealed a distorted T-shaped geometry like the common hypervalent λ^3-iodanes with an O11–I10–N16 bond angle of 162.32(8)° and I–ONO₂ bond length of 2.336(2)Å. The length of the observed I–ONO₂ bond is significantly longer than its average I–O bond length of 2.14^[34] and longer than the I–ONO₂ bond length in the analogous O₂NO-containing I(III) 1a and 1c, i.e., 2.311(3)Å,^[16] and 2.283(2)Å^[13] respectively, suggesting reduced covalent character. Compound 1d also possesses a planar geometry, as indicated by the torsion angles O14–N13–O11–I10 (8.7(3)°), I10–C19–C18–C20 (2.0(3)°) and O17–C20–N16–C21 (8.0(5)°).	showed that compound 1d decomposed at 180.7 °C (see SI for details). Furthermore, a single crystal of 1d was grown in a mixed solvents of chloroform/n-hexane at room temperature, and it crystallized in the monoclinic space group $P2_1/c$ with $Z = 4$. A X-ray crystal analysis of compound 1d (Scheme 1) revealed a distorted T-shaped geometry like the common hypervalent λ^3-iodanes with an O11–I10–N16 bond angle of 162.32(8)° and I–ONO₂ bond length of 2.336(2) Å. The length of the observed I–ONO₂ bond in compound 1d is longer than its analogous 1a and 1c, i.e., 2.311(3) Å,^[16] and 2.283(2) Å^[13] respectively, suggesting reduced covalent character. Compound 1d also possesses a planar geometry, as indicated by the torsion angles O14–N13–O11–I10 (8.7(3)°), I10–C19–C18–C20 (2.0(3)°) and O17–C20–N16–C21 (8.0(5)°).
Page 2, Paragraph 4	Other metal additives including FeBr₂, PdCl₂, Mn(OAc)₂, Ni(acac)₂, Co(acac)₂ and RhCl(PPh₃)₃ were also investigated, but none was proved to be compatible with this reaction (see SI for details). The result of a control reaction conducted in the absence of copper catalyst provided no desired product, indicating that the copper catalyst is indispensable for the reaction to occur (Table 1, entry 14). Temperature was proved to be another important factor for an efficient transformation, with reaction run at 60 °C afforded the best outcome (Table 1, entries 15-17). Furthermore, the screening on dosage of O₂NO-I(III) 1d indicated that 1.5 equivalents of the hypervalent iodine(III) reagent was necessary for a complete consumption of the starting enamine 2a (Table 1, entry 18).	Other metal additives including FeBr₂, PdCl₂, Mn(OAc)₂, Ni(acac)₂, Co(acac)₂ and RhCl(PPh₃)₃ were also investigated. All of them were proved to be compatible with this reaction except Co(acac)₂ (see SI for details). The result of a control reaction conducted in the absence of copper catalyst provided no desired product, indicating that the copper catalyst is indispensable for the reaction to occur (Table 1, entry 14). Temperature was proved to be another important factor for an efficient transformation, with reaction run at 60 °C afforded the best outcome (Table 1, entries 15-17). Furthermore, the screening on dosage of O₂NO-I(III) 1d indicated that 1.5 equivalents of the hypervalent iodine(III) reagent was necessary for a complete consumption of the starting enamine 2a (Table 1, entries 18-19).

Page 2, Table 1	    1d Entry(x equiv) Catalyst Solvent T (°C) Yield (%)^[b]    11.5CuIMeCN5072 21.5CuIDCE5043 31.5CuI1,4- dioxane5059 41.5CuITHF50nd 51.5CuIDMF50nd 61.5CuIHFIP50nd 71.5CuBrMeCN5066 81.5CuSCNMeCN5063 91.5CuClMeCN5053 101.5Cu₂OMeCN5058 111.5CuBr₂MeCN5060 121.5Cu(OAc)₂MeCN5031 131.5Cu(OTf)₂MeCN5047 141.5noneMeCN50nd 151.5CuIMeCNrttrace 161.5CuIMeCN3022 171.5CuIMeCN6080 181.0CuIMeCN6064  		1d Entry(x equiv)	Catalyst	Solvent	T (°C)	Yield (%) ^[b]	1	1.5	CuI	MeCN	50	72	2	1.5	CuI	DCE	50	43	3	1.5	CuI	1,4- dioxane	50	59	4	1.5	CuI	THF	50	nd	5	1.5	CuI	DMF	50	nd	6	1.5	CuI	HFIP	50	nd	7	1.5	CuBr	MeCN	50	66	8	1.5	CuSCN	MeCN	50	63	9	1.5	CuCl	MeCN	50	53	10	1.5	Cu ₂ O	MeCN	50	58	11	1.5	CuBr ₂	MeCN	50	60	12	1.5	Cu(OAc) ₂	MeCN	50	31	13	1.5	Cu(OTf) ₂	MeCN	50	47	14	1.5	none	MeCN	50	nd	15	1.5	CuI	MeCN	rt	trace	16	1.5	CuI	MeCN	30	22	17	1.5	CuI	MeCN	60	80	18	1.0	CuI	MeCN	60	64	    1d Entry(x equiv) Catalyst Solvent T (°C) Yield (%)^[b]    11.5CuIMeCN5072 21.5CuIDCE5043 31.5CuI1,4- dioxane5059 41.5CuITHF50nd 51.5CuIDMF50nd 61.5CuIHFIP50nd 71.5CuBrMeCN5066 81.5CuSCNMeCN5063 91.5CuClMeCN5053 101.5Cu₂OMeCN5058 111.5CuBr₂MeCN5060 121.5Cu(OAc)₂MeCN5031 131.5Cu(OTf)₂MeCN5047 141.5noneMeCN50nd 151.5CuIMeCNrttrace 161.5CuIMeCN3022 171.5CuIMeCN6080 181.25CuIMeCN6073 191.0CuIMeCN6064  		1d Entry(x equiv)	Catalyst	Solvent	T (°C)	Yield (%) ^[b]	1	1.5	CuI	MeCN	50	72	2	1.5	CuI	DCE	50	43	3	1.5	CuI	1,4- dioxane	50	59	4	1.5	CuI	THF	50	nd	5	1.5	CuI	DMF	50	nd	6	1.5	CuI	HFIP	50	nd	7	1.5	CuBr	MeCN	50	66	8	1.5	CuSCN	MeCN	50	63	9	1.5	CuCl	MeCN	50	53	10	1.5	Cu ₂ O	MeCN	50	58	11	1.5	CuBr ₂	MeCN	50	60	12	1.5	Cu(OAc)₂	MeCN	50	31	13	1.5	Cu(OTf) ₂	MeCN	50	47	14	1.5	none	MeCN	50	nd	15	1.5	CuI	MeCN	rt	trace	16	1.5	CuI	MeCN	30	22	17	1.5	CuI	MeCN	60	80	18	1.25	CuI	MeCN	60	73	19	1.0	CuI	MeCN	60	64
	1d Entry(x equiv)	Catalyst	Solvent	T (°C)	Yield (%) ^[b]																																																																																																																																																																																																																																							
1	1.5	CuI	MeCN	50	72																																																																																																																																																																																																																																							
2	1.5	CuI	DCE	50	43																																																																																																																																																																																																																																							
3	1.5	CuI	1,4- dioxane	50	59																																																																																																																																																																																																																																							
4	1.5	CuI	THF	50	nd																																																																																																																																																																																																																																							
5	1.5	CuI	DMF	50	nd																																																																																																																																																																																																																																							
6	1.5	CuI	HFIP	50	nd																																																																																																																																																																																																																																							
7	1.5	CuBr	MeCN	50	66																																																																																																																																																																																																																																							
8	1.5	CuSCN	MeCN	50	63																																																																																																																																																																																																																																							
9	1.5	CuCl	MeCN	50	53																																																																																																																																																																																																																																							
10	1.5	Cu ₂ O	MeCN	50	58																																																																																																																																																																																																																																							
11	1.5	CuBr ₂	MeCN	50	60																																																																																																																																																																																																																																							
12	1.5	Cu(OAc) ₂	MeCN	50	31																																																																																																																																																																																																																																							
13	1.5	Cu(OTf) ₂	MeCN	50	47																																																																																																																																																																																																																																							
14	1.5	none	MeCN	50	nd																																																																																																																																																																																																																																							
15	1.5	CuI	MeCN	rt	trace																																																																																																																																																																																																																																							
16	1.5	CuI	MeCN	30	22																																																																																																																																																																																																																																							
17	1.5	CuI	MeCN	60	80																																																																																																																																																																																																																																							
18	1.0	CuI	MeCN	60	64																																																																																																																																																																																																																																							
	1d Entry(x equiv)	Catalyst	Solvent	T (°C)	Yield (%) ^[b]																																																																																																																																																																																																																																							
1	1.5	CuI	MeCN	50	72																																																																																																																																																																																																																																							
2	1.5	CuI	DCE	50	43																																																																																																																																																																																																																																							
3	1.5	CuI	1,4- dioxane	50	59																																																																																																																																																																																																																																							
4	1.5	CuI	THF	50	nd																																																																																																																																																																																																																																							
5	1.5	CuI	DMF	50	nd																																																																																																																																																																																																																																							
6	1.5	CuI	HFIP	50	nd																																																																																																																																																																																																																																							
7	1.5	CuBr	MeCN	50	66																																																																																																																																																																																																																																							
8	1.5	CuSCN	MeCN	50	63																																																																																																																																																																																																																																							
9	1.5	CuCl	MeCN	50	53																																																																																																																																																																																																																																							
10	1.5	Cu ₂ O	MeCN	50	58																																																																																																																																																																																																																																							
11	1.5	CuBr ₂	MeCN	50	60																																																																																																																																																																																																																																							
12	1.5	Cu(OAc)₂	MeCN	50	31																																																																																																																																																																																																																																							
13	1.5	Cu(OTf) ₂	MeCN	50	47																																																																																																																																																																																																																																							
14	1.5	none	MeCN	50	nd																																																																																																																																																																																																																																							
15	1.5	CuI	MeCN	rt	trace																																																																																																																																																																																																																																							
16	1.5	CuI	MeCN	30	22																																																																																																																																																																																																																																							
17	1.5	CuI	MeCN	60	80																																																																																																																																																																																																																																							
18	1.25	CuI	MeCN	60	73																																																																																																																																																																																																																																							
19	1.0	CuI	MeCN	60	64																																																																																																																																																																																																																																							
Page 3, Paragraph 2	To our surprise, when the reaction of substrate 4a with a menaphthyl moiety was conducted under above optimized reaction conditions, it was not the expected menaphthyl furazan but naphthoyl furazan 5a as well as its precursor 5a' were isolated in a yield of 23% and 55%, respectively (Table 3, entry 1).	To our surprise, when the reaction of substrate 4a with a menaphthyl moiety was conducted under above optimized reaction conditions, it was not the expected menaphthyl furazan but the benzylic CH₂-oxidized compounds, i.e., naphthoyl furazan 5a as well as its precursor 5a' that were isolated in a yield of 23% and 55%, respectively (Table 3, entry 1).																																																																																																																																																																																																																																										
Page 3, Paragraph 3	Derivatization of the obtained furazan derivatives were carried out to prove the utility of this method (Scheme 2). To our delight, furazan 3b could be further transformed into compound 7 via the one-pot two-step amidation. ^[35] In addition, azide 6 could be achieved from furazan 3k by reacting with methyl glycinate through aminolysis of ester. ^[36]	Derivatization of the obtained furazan derivatives were carried out to prove the utility of this method (Scheme 2). To our delight, furazan 3b could be further transformed into compound 6 via the one-pot two-step amidation. ^[34] In addition, azide 7 could be achieved from furazan 3k by reacting with methyl glycinate through aminolysis of ester. ^[35]																																																																																																																																																																																																																																										

Scheme 5. DFT-Computed potential energy profile for the reaction of imine **D** to access the cyclization/dehydration product **3a** under the standard conditions (standard state: 25°C, 1 mol/L) at RI-PBWPB95-D3(BJ)/def2-QZVPP-SMD-(Acetonitrile)//B3LYP-D3(BJ)/def2-SVP-SMD-(Acetonitrile) level of theory.

delete

In order to well explain the transformation of imine **D** into furazan **3a**, two pathways (Scheme 4, path a and b) were postulated. Path a encompasses the first conversion of imine **D** to enamine **F**, then the formation of cyclized intermediate **G** from the nucleophilic attack of nitrogen atom of enamino moiety to oxygen center of nitrene in intermediate **F**, and the subsequent tautomerization (via intermediates **H-J**) and dehydration to give product **3a**. While in path b, the intramolecular attack of oxygen atom of nitrene to nitrogen center of imino moiety in intermediate **D**, with the concomitant formation of C-C double bond and cleavage of C-N bond occurs first to give intermediate **K**. Then intramolecular cyclization of **K** provides the cyclized intermediate **L**, which undergoes similar tautomerization (via intermediates **M-J**) and dehydration to afford furazan **3a**.

Next, two pathways (Scheme 4, path a and b) were postulated for the formation of furazan **3a** from intermediate **D**. In path a, enamine **E** was formed from imine **D** via tautomerism first. Then nucleophilic attack of nitrogen atom of enamino moiety in intermediate **E** to its oxygen center of nitrene gave the cyclized intermediate **F**. Subsequent tautomerization of **F** achieved via the system of **S1/S2** and following dehydration of the resulting intermediate **H** gave product **3a**. While in path b, the intramolecular attack of oxygen atom of nitrene in intermediate **D** to nitrogen center of its imino moiety, with the concomitant formation of C-C double bond and cleavage of C-N bond occurred first to give intermediate **I**. Then intramolecular cyclization of **I** provided the cyclized intermediate **J**,

	To gain more evidence for the most possible mechanistic pathway, we performed density functional theory (DFT) calculations on the reaction of imine D under standard conditions (Scheme 5). As can be seen from the calculation results, in path a, D first bounds with S1 to form intermediate CP, which undergoes deprotonation to form intermediate R-E, a resonance species of E, through TSCP-E with free energy barrier of 24.3 kcal/mol. Further a protonation occurs at nitrogen atom of R-E to produce the cyclized precursor F. Then intramolecular cyclization of F affords intermediate G through the transition state TSF-G with a high energy barrier of 87.3 kcal/mol. Subsequently, a series of deprotonation-protonation reactions of intermediate G and the final dehydration give the final product 3a. Comparatively in path b, D first undergoes cyclization to produce intermediate L through transition state TSD-L, with a relatively lower free energy barrier of 66.8 kcal/mol. Notably, this step occurs not through the imaged intermediate K drawn in Scheme 4. Then intermediate J is obtained after several deprotonation-protonation reactions of L. Judged by the free energy barrier, path b should be a more favorable mechanistic pathway for the cyclization process.	which underwent similar tautomerization and following dehydration of the resulting intermediate H to afford furazan 3a.
Page 8, References	34. Kiprof, P. The Nature of Iodine Oxygen Bonds in Hypervalent 10-1-3 Iodine Compounds. Arkivoc 4 , 19-25 (2005).	delete
Page 8-9, References	35. Yang, Z.-F., Xu, C., Zheng, X. & Zhang, X. Nickel-catalyzed Carbodifunctionalization of N-Vinylamides Enables Access to γ-Amino Acids. Chem. Commun. 56, 2642-2645 (2020). 36. (a) Yang, X., Wang, Z., Fang, X., Yang, X., Wu, F. & Shen, Y. Synthesis of Difluoromethylene-containing 1,2,4-Oxadiazole Compounds via the Reaction of 5-(Difluoroiodomethyl)-3-phenyl-1,2,4-oxadiazole with Unsaturated Compounds Initiated by Sodium Dithionite. Synthesis 12, 1768-1778 (2007); (b) Lamarque, J.-F., Lamarque, C., Lassara, S., Médebielle, M., Molette, J., David, E., Pellet-Rostaing, S., Lemaire, M., Okada, E., Shibata, D. & Pilet, G. Copper Catalyzed 1,3-Dipolar Cycloaddition Reaction of Azides with N-(2-Trifluoroacetylaryl)	34. Yang, Z.-F., Xu, C., Zheng, X. & Zhang, X. Nickel-catalyzed Carbodifunctionalization of N-Vinylamides Enables Access to γ-Amino Acids. Chem. Commun. 56, 2642-2645 (2020). 35. (a) Yang, X., Wang, Z., Fang, X., Yang, X., Wu, F. & Shen, Y. Synthesis of Difluoromethylene-containing 1,2,4-Oxadiazole Compounds via the Reaction of 5-(Difluoroiodomethyl)-3-phenyl-1,2,4-oxadiazole with Unsaturated Compounds Initiated by Sodium Dithionite. Synthesis 12, 1768-1778 (2007); (b) Lamarque, J.-F., Lamarque, C., Lassara, S., Médebielle, M., Molette, J., David, E., Pellet-Rostaing, S., Lemaire, M., Okada, E., Shibata, D. & Pilet, G. Copper Catalyzed 1,3-Dipolar Cycloaddition Reaction of Azides with N-(2-Trifluoroacetylaryl)

	Propargylamines: A Mild Entry to Novel 1,4-Disubstituted-[1,2,3]-triazole Derivatives. J. Fluorine Chem. 129, 788-798 (2008). 37. (a) Maity, S., Manna, S., Rana, S., Naveen, T., Mallick, A. & Maiti, D. Efficient and Stereoselective Nitration of Mono-and Disubstituted Olefins with AgNO₂ and TEMPO. J. Am. Chem. Soc. 135, 3355-3358 (2013); (b) Fan, Z., Ni, J. & Zhang, A. Meta-selective C_{Ar}-H Nitration of Arenes Through a Ru₃(CO)₁₂-catalyzed Ortho-metalation Strategy. J. Am. Chem. Soc. 138, 8470-8475 (2016); (c) Parrino, F., Livraghi, S., Giamello, E. & Palmisano, L. The Existence of Nitrate Radicals in Irradiated TiO₂ Aqueous Suspensions in the Presence of Nitrate Ions. Angew. Chem. Int. Ed. 57, 10702-10706 (2018); (d) Huang, J., Ding, F., Rojsitthisak, P., He, F.-S. & Wu, J. Recent Advances in Nitro-involved Radical Reactions. Org. Chem. Front. 7, 2873-2898 (2020). 38. (a) Kirovskaya, I. A., Mironova, E. V., Bykova, E. I., Timoshenko, O. T. & Filatova, T. N. Adsorption and Electrophysical Studies of the Sensitivity and Selectivity of the Surface of the InSb-CdTe System with Respect to Toxic Gases. Russ. J. Phys. Chem. 82, 830-834 (2008); (b) Rilyanti, M. & Hadi, S. Synthesis, Characterization and Thermal Stability of Complex Cis-[Co(bipy)₂(CN)₂] and Its Interaction with NO₂ Gas. Russ. J. Inorg. Chem. 56, 418-421 (2011). 39. (a) Li, Y., Gao, L.-X. & Han, F.-S. Reliable and Diverse Synthesis of Aryl Azides Through Copper-catalyzed Coupling of Boronic Acids or Esters with TMSN₃. Chem. Eur. J. 16, 7969-7972 (2010); (b) Wang, Y., Li, G.-X., Yang, G., He, G. & Chen, G. A Visible-light-promoted Radical Reaction System for Azidation and Halogenation of Tertiary aliphatic C-H bonds. Chem. Sci. 7, 2679-2683 (2016); (c) Rabet, P. T. G., Fumagalli, G., Boyd, S. & Greaney, M. F. Benzylic C-H Azidation Using the Zhdankin Reagent and a Copper Photoredox Catalyst. Org. Lett. 18, 1646-1649 (2016); (d) Shinomoto, Y., Yoshimura, A., Shimizu, H., Yamazaki, M., Zhdankin, V. V. & Saito, A. Tetra-n-butylammonium Iodide Catalyzed C-H Azidation of Aldehydes with Thermally Stable Azidobenziodoxolone. Org. Lett. 17, 5212-5215 (2015); (e) Muriel, B. & Waser, J. Azide Radical	Propargylamines: A Mild Entry to Novel 1,4-Disubstituted-[1,2,3]-triazole Derivatives. J. Fluorine Chem. 129, 788-798 (2008). 36. (a) Maity, S., Manna, S., Rana, S., Naveen, T., Mallick, A. & Maiti, D. Efficient and Stereoselective Nitration of Mono-and Disubstituted Olefins with AgNO₂ and TEMPO. J. Am. Chem. Soc. 135, 3355-3358 (2013); (b) Fan, Z., Ni, J. & Zhang, A. Meta-selective C_{Ar}-H Nitration of Arenes Through a Ru₃(CO)₁₂-catalyzed Ortho-metalation Strategy. J. Am. Chem. Soc. 138, 8470-8475 (2016); (c) Parrino, F., Livraghi, S., Giamello, E. & Palmisano, L. The Existence of Nitrate Radicals in Irradiated TiO₂ Aqueous Suspensions in the Presence of Nitrate Ions. Angew. Chem. Int. Ed. 57, 10702-10706 (2018); (d) Huang, J., Ding, F., Rojsitthisak, P., He, F.-S. & Wu, J. Recent Advances in Nitro-involved Radical Reactions. Org. Chem. Front. 7, 2873-2898 (2020). 37. (a) Kirovskaya, I. A., Mironova, E. V., Bykova, E. I., Timoshenko, O. T. & Filatova, T. N. Adsorption and Electrophysical Studies of the Sensitivity and Selectivity of the Surface of the InSb-CdTe System with Respect to Toxic Gases. Russ. J. Phys. Chem. 82, 830-834 (2008); (b) Rilyanti, M. & Hadi, S. Synthesis, Characterization and Thermal Stability of Complex Cis-[Co(bipy)₂(CN)₂] and Its Interaction with NO₂ Gas. Russ. J. Inorg. Chem. 56, 418-421 (2011). 38. (a) Li, Y., Gao, L.-X. & Han, F.-S. Reliable and Diverse Synthesis of Aryl Azides Through Copper-catalyzed Coupling of Boronic Acids or Esters with TMSN₃. Chem. Eur. J. 16, 7969-7972 (2010); (b) Wang, Y., Li, G.-X., Yang, G., He, G. & Chen, G. A Visible-light-promoted Radical Reaction System for Azidation and Halogenation of Tertiary aliphatic C-H bonds. Chem. Sci. 7, 2679-2683 (2016); (c) Rabet, P. T. G., Fumagalli, G., Boyd, S. & Greaney, M. F. Benzylic C-H Azidation Using the Zhdankin Reagent and a Copper Photoredox Catalyst. Org. Lett. 18, 1646-1649 (2016); (d) Shinomoto, Y., Yoshimura, A., Shimizu, H., Yamazaki, M., Zhdankin, V. V. & Saito, A. Tetra-n-butylammonium Iodide Catalyzed C-H Azidation of Aldehydes with Thermally Stable Azidobenziodoxolone. Org. Lett. 17, 5212-5215 (2015); (e) Muriel, B. & Waser, J. Azide Radical Initiated Ring Opening of Cyclopropenes Leading to Alkenyl
--	---	--

	Initiated Ring Opening of Cyclopropenes Leading to Alkenyl Nitriles and Polycyclic Aromatic Compounds. Angew. Chem. Int. Ed. 60, 4075-4079 (2021); (f) Matsumoto, K., Nakajima, M. & Nemoto, T. Determination of the Best Functional and Basis Sets for Optimization of the Structure of Hypervalent Iodines and Calculation of Their First and Second Bond Dissociation Enthalpies. J. Phys. Org. Chem. 32, e3961 (2019).	Nitriles and Polycyclic Aromatic Compounds. Angew. Chem. Int. Ed. 60, 4075-4079 (2021); (f) Matsumoto, K., Nakajima, M. & Nemoto, T. Determination of the Best Functional and Basis Sets for Optimization of the Structure of Hypervalent Iodines and Calculation of Their First and Second Bond Dissociation Enthalpies. J. Phys. Org. Chem. 32, e3961 (2019).
Page 9, Author Contributions	Z. Y. and Y. D. conceived and designed the experiments; Z. F. carried out most of experiments; C. X. carried out the DFT calculation; J. X. carried out the single-crystal X-ray experiments; Y. S., X. L. and B. J. analyzed data; Z. Y. and Y. D. directed the project and wrote the paper.	Z. Y. and Y. D. conceived and designed the experiments; Z. F. carried out most of experiments; J. X. carried out the single-crystal X-ray experiments; Y. S., X. L. and B. J. analyzed data; Z. Y. and Y. D. directed the project and wrote the paper.

The end

REVIEWERS' COMMENTS:

Reviewer #2 (Remarks to the Author):

The authors have satisfyingly answered the concern of the reviewers and acceptance is recommended now.

Reviewer #3 (Remarks to the Author):

Thank you for addressing all the points. I confirm that this article is suitable for Communications Chemistry.

Reviewer #4 (Remarks to the Author):

The authors have decided to remove the DFT part of the previous manuscript version. I do think this is a good decision as the proper computational mechanistic study is very complex and probably outside of the scope of the manuscript. As I pointed out in my previous revision, I think the manuscript is interesting and well-executed from the experimental point of view (both the methodology development and the experimental mechanistic studies), so I support publication of the manuscript in this revised version.

REVIEWERS' COMMENTS

Reviewer #2 (Remarks to the Author):

The authors have satisfyingly answered the concern of the reviewers and acceptance is recommended now.

Response: We greatly appreciate the reviewer for this positive comment. We sincerely thank the reviewer for supporting the publication of our manuscript.

Reviewer #3 (Remarks to the Author):

Thank you for addressing all the points. I confirm that this article is suitable for Communications Chemistry.

Response: We greatly appreciate the reviewer for this positive comment. We sincerely thank the reviewer for supporting the publication of our manuscript.

Reviewer #4 (Remarks to the Author):

The authors have decided to remove the DFT part of the previous manuscript version. I do think this is a good decision as the proper computational mechanistic study is very complex and probably outside of the scope of the manuscript. As I pointed out in my previous revision, I think the manuscript is interesting and well-executed from the experimental point of view (both the methodology development and the experimental mechanistic studies), so I support publication of the manuscript in this revised version.

Response: We greatly appreciate the reviewer for this positive comment. We sincerely thank the reviewer for supporting the publication of our manuscript.

Table. A Point-to-point Description of Changes

Location	Original MS	Revised MS
Page 1 Title	Concomitant 1,2-Aryl Migration/Elimination Reaction Mediated by Hypervalent Iodine Reagents: Chemoselective Cycloisomerization of O -Alkenylbenzamides	Chemoselective Cycloisomerization of O -alkenylbenzamides via Concomitant 1,2-Aryl migration/Elimination Mediated by Hypervalent Iodine Reagents
Page 1 Author name	Jiixin He ^{1,3} , Feng-Huan Du ^{2,3} , Chi Zhang ^{2,*} & Yunfei Du ^{1,*}	Jiixin He ¹ , Feng-Huan Du ² , Chi Zhang ^{2,*} & Yunfei Du ^{1,*}
Page 1 Author tagging statements	³ These authors contributed equally.	³ These authors contributed equally: Jiixin He, Feng-Huan Du.
Page 1 Corresponding author's email	E-mail: duyunfeier@tju.edu.cn; zhangchi@nankai.edu.cn.	e-mail: zhangchi@nankai.edu.cn; duyunfeier@tju.edu.cn.
Page 1 Abstract	A chemodivergent cycloisomerization approach to construct isoquinolinone and iminoisocoumarin skeletons from o -alkenylbenzamide derivatives is established. The chemo-controllable strategy employed an exclusive 1,2-aryl migration/elimination cascade, enabled by different hypervalent iodine species generated in situ from the reaction of iodosobenzene (PhIO) with MeOH or 2,4,6-tris-isopropylbenzene sulfonic acid. DFT studies revealed that the nitrogen and oxygen atoms of the intermediates in the two reaction systems have different nucleophilicities and thus produce the selectivity of N or O -attack modes.	As an ambident nucleophile, controlling the reaction selectivities of nitrogen and oxygen atoms in amide moiety is a challenging issue in organic synthesis. Herein, a chemodivergent cycloisomerization approach to construct isoquinolinone and iminoisocoumarin skeletons from o-alkenylbenzamide derivatives is established. The chemo-controllable strategy employed an exclusive 1,2-aryl migration/elimination cascade, enabled by different hypervalent iodine species generated in situ from the reaction of iodosobenzene (PhIO) with MeOH or 2,4,6-tris-isopropylbenzene sulfonic acid. DFT studies revealed that the nitrogen and oxygen atoms of the intermediates in the two reaction systems have different nucleophilicities and thus produce the selectivity of N or O-attack modes.
Page 1	added heading	Introduction
Page 1 Right column Paragraph 2	Herein, we present our results in controlling the reactivity of o -alkenylbenzamides by hypervalent iodine species generated in situ .	Herein, we present our results in controlling the reactivity of o -alkenylbenzamides by hypervalent iodine species generated in situ .
Page 1 Left column Paragraph 1	In this regard, a new protocol dictating the chemodivergent cycloisomerization of o -alkenylbenzamides, by tuning the differentiation of the N vs O nucleophilic strength, to construct novel heterocyclic skeletons should be highly desirable.	In this regard, a new protocol dictating the chemodivergent cycloisomerization of o -alkenylbenzamides, by tuning the differentiation of the N vs O nucleophilic strength, to construct novel heterocyclic skeletons should be highly desirable.
Page 1 Paragraph 2	scheme 1a; scheme 1b; scheme 1c	Fig. 1a; Fig. 1b; Fig. 1c

Page 1 Right column	Scheme 1. Strategies for divergent cyclization of amide derivatives.	Fig. 1 Strategies for divergent cyclization of amide derivatives. a,b Previous works on cycloisomerization of amide derivatives. c This work, hypervalent iodine reagents mediated intramolecular cycloisomerization and 1,2-aryl migration.																																																																																																																																																																																																																																																																																																						
Page 1 Right column Paragraph 3	When substrate 1a was treated with iodosobenzene (PhIO) in MeOH, combined with BF ₃ ·OEt ₂ (0.2 equiv) as a Lewis acid, we found the reaction displayed a completely distinct N -attack cyclization mode to furnish isoquinolinone 2a in 56% yield.	When substrate 1a was treated with iodosobenzene (PhIO) in MeOH, combined with BF ₃ ·OEt ₂ (0.2 equiv) as a Lewis acid, we found the reaction displayed a completely distinct N -attack cyclization mode to furnish isoquinolinone 2a in 56% yield.																																																																																																																																																																																																																																																																																																						
Page 2 Left column Table 1	^[a] Reaction conditions: 1a (0.5 mmol), HIR (1.5 equiv), solvent (5.0 mL), rt. NR = no reaction. ^[b] Entries 3-12, additives (1.5 equiv) were used; entries 13-20, additives (0.2 equiv) were used. ^[c] Isolated yield. ^[d] BF ₃ ·OEt ₂ (1.5 equiv) was added. ^[e] TMSOTf (1.5 equiv) was added. ^[f] LiClO ₄ (1.5 equiv) was added. ^[g] Zn(ClO ₄) ₂ (1.5 equiv) was added. ^[h] LiClO ₄ (1.0 equiv) was added. ^[i] LiClO ₄ (1.8 equiv) was added. See SI For more details.	^{a)} Reaction conditions: 1a (0.5 mmol), HIR (1.5 equiv), solvent (5.0 mL), rt. NR = no reaction. ^[b] Entries 3-12, additives (1.5 equiv) were used; entries 13-20, additives (0.2 equiv) were used. ^[c] Isolated yield. ^[d] BF ₃ ·OEt ₂ (1.5 equiv) was added. ^[e] TMSOTf (1.5 equiv) was added. ^[f] LiClO ₄ (1.5 equiv) was added. ^[g] Zn(ClO ₄) ₂ (1.5 equiv) was added. ^[h] LiClO ₄ (1.0 equiv) was added. ^[i] LiClO ₄ (1.8 equiv) was added. (For details see Supplementary Table S1, S2).																																																																																																																																																																																																																																																																																																						
Page 2 Left column Paragraph 4	Encouraged by the above results, we then turned our attention toward exploring the chemodivergent pattern of the protocol for the synthesis of O -cyclized iminoisocoumarin products.	Encouraged by the above results, we then turned our attention toward exploring the chemodivergent pattern of the protocol for the synthesis of O -cyclized iminoisocoumarin products.																																																																																																																																																																																																																																																																																																						
Page 2 Left column Table 1	   Entry^{a)} HIR^{b)} Solvent^{c)} Additive^{d)} T (°C)^{e)} Yield^{f)} of 2a (%) Yield^{f)} of 3a (%)    1^{c)}PhIO^{c)}MeOH^{c)}BF₃·OEt₂^{c)}rt^{c)}56^{c)}—^{c)} 2^{c)}PhIO^{c)}MeOH^{c)}Et₃N^{c)}rt^{c)}NR^{c)}—^{c)} 3^{c)}PhIO^{c)}MeOH^{c)}TFA^{c)}rt^{c)}61^{c)}—^{c)} 4^{c)}PhIO^{c)}MeOH^{c)}TfOH^{c)}rt^{c)}63^{c)}—^{c)} 5^{c)}PhIO^{c)}MeOH^{c)}TMSOTf^{c)}rt^{c)}70^{c)}—^{c)} 6^{c)}PhIO^{c)}MeOH^{c)}LiClO₄^{c)}rt^{c)}NR^{c)}—^{c)} 7^{c)}PhIO^{c)}MeOH^{c)}50% H₂SO₄^{c)}rt^{c)}58^{c)}—^{c)} 8^{c)}PhIO^{c)}MeOH^{c)}TMSOTf^{c)}reflux^{c)}81^{c)}—^{c)} 9^{c)}HTIB^{c)}DCE^{c)}—^{c)}rt^{c)}—^{c)}44^{c)} 10^{c)}HTIB^{c)}DCE^{c)}—^{c)}80^{c)}—^{c)}55^{c)} 11^{c)}PhIO^{c)}DCE^{c)}S1^{c)}80^{c)}—^{c)}54^{c)} 12^{c)}PhIO^{c)}DCE^{c)}S2^{c)}80^{c)}—^{c)}52^{c)} 13^{c)}PhIO^{c)}DCE^{c)}S3^{c)}80^{c)}—^{c)}63^{c)} 14^{c)}PhIO^{c)}DCE^{c)}S4^{c)}80^{c)}—^{c)}69^{c)} 15^{d)}PhIO^{c)}DCE^{c)}S4^{c)}80^{c)}—^{c)}73^{c)} 16^{d)}PhIO^{c)}DCE^{c)}S4^{c)}80^{c)}—^{c)}70^{c)} 17^{d)}PhIO^{c)}DCE^{c)}S4^{c)}80^{c)}—^{c)}90^{c)} 18^{d)}PhIO^{c)}DCE^{c)}S4^{c)}80^{c)}—^{c)}82^{c)} 19^{d)}PhIO^{c)}DCE^{c)}S4^{c)}80^{c)}—^{c)}77^{c)} 20^{d)}PhIO^{c)}DCE^{c)}S4^{c)}80^{c)}—^{c)}90^{c)}  	Entry ^{a)}	HIR ^{b)}	Solvent ^{c)}	Additive ^{d)}	T (°C) ^{e)}	Yield ^{f)} of 2a (%)	Yield ^{f)} of 3a (%)	1 ^{c)}	PhIO ^{c)}	MeOH ^{c)}	BF ₃ ·OEt ₂ ^{c)}	rt ^{c)}	56 ^{c)}	— ^{c)}	2 ^{c)}	PhIO ^{c)}	MeOH ^{c)}	Et ₃ N ^{c)}	rt ^{c)}	NR ^{c)}	— ^{c)}	3 ^{c)}	PhIO ^{c)}	MeOH ^{c)}	TFA ^{c)}	rt ^{c)}	61 ^{c)}	— ^{c)}	4 ^{c)}	PhIO ^{c)}	MeOH ^{c)}	TfOH ^{c)}	rt ^{c)}	63 ^{c)}	— ^{c)}	5 ^{c)}	PhIO ^{c)}	MeOH ^{c)}	TMSOTf ^{c)}	rt ^{c)}	70 ^{c)}	— ^{c)}	6 ^{c)}	PhIO ^{c)}	MeOH ^{c)}	LiClO ₄ ^{c)}	rt ^{c)}	NR ^{c)}	— ^{c)}	7 ^{c)}	PhIO ^{c)}	MeOH ^{c)}	50% H ₂ SO ₄ ^{c)}	rt ^{c)}	58 ^{c)}	— ^{c)}	8 ^{c)}	PhIO ^{c)}	MeOH ^{c)}	TMSOTf ^{c)}	reflux ^{c)}	81 ^{c)}	— ^{c)}	9 ^{c)}	HTIB ^{c)}	DCE ^{c)}	— ^{c)}	rt ^{c)}	— ^{c)}	44 ^{c)}	10 ^{c)}	HTIB ^{c)}	DCE ^{c)}	— ^{c)}	80 ^{c)}	— ^{c)}	55 ^{c)}	11 ^{c)}	PhIO ^{c)}	DCE ^{c)}	S1 ^{c)}	80 ^{c)}	— ^{c)}	54 ^{c)}	12 ^{c)}	PhIO ^{c)}	DCE ^{c)}	S2 ^{c)}	80 ^{c)}	— ^{c)}	52 ^{c)}	13 ^{c)}	PhIO ^{c)}	DCE ^{c)}	S3 ^{c)}	80 ^{c)}	— ^{c)}	63 ^{c)}	14 ^{c)}	PhIO ^{c)}	DCE ^{c)}	S4 ^{c)}	80 ^{c)}	— ^{c)}	69 ^{c)}	15 ^{d)}	PhIO ^{c)}	DCE ^{c)}	S4 ^{c)}	80 ^{c)}	— ^{c)}	73 ^{c)}	16 ^{d)}	PhIO ^{c)}	DCE ^{c)}	S4 ^{c)}	80 ^{c)}	— ^{c)}	70 ^{c)}	17 ^{d)}	PhIO ^{c)}	DCE ^{c)}	S4 ^{c)}	80 ^{c)}	— ^{c)}	90 ^{c)}	18 ^{d)}	PhIO ^{c)}	DCE ^{c)}	S4 ^{c)}	80 ^{c)}	— ^{c)}	82 ^{c)}	19 ^{d)}	PhIO ^{c)}	DCE ^{c)}	S4 ^{c)}	80 ^{c)}	— ^{c)}	77 ^{c)}	20 ^{d)}	PhIO ^{c)}	DCE ^{c)}	S4 ^{c)}	80 ^{c)}	— ^{c)}	90 ^{c)}	   Entry^{a)} HIR^{b)} Solvent^{c)} Additive^{d)} T (°C)^{e)} Yield^{f)} of 2a (%) Yield^{f)} of 3a (%)    1^{c)}PhIO^{c)}MeOH^{c)}BF₃·OEt₂^{c)}rt^{c)}56^{c)}—^{c)} 2^{c)}PhIO^{c)}MeOH^{c)}Et₃N^{c)}rt^{c)}NR^{c)}—^{c)} 3^{c)}PhIO^{c)}MeOH^{c)}TFA^{c)}rt^{c)}61^{c)}—^{c)} 4^{c)}PhIO^{c)}MeOH^{c)}TfOH^{c)}rt^{c)}63^{c)}—^{c)} 5^{c)}PhIO^{c)}MeOH^{c)}TMSOTf^{c)}rt^{c)}70^{c)}—^{c)} 6^{c)}PhIO^{c)}MeOH^{c)}LiClO₄^{c)}rt^{c)}NR^{c)}—^{c)} 7^{c)}PhIO^{c)}MeOH^{c)}50% H₂SO₄^{c)}rt^{c)}58^{c)}—^{c)} 8^{c)}PhIO^{c)}MeOH^{c)}TMSOTf^{c)}reflux^{c)}81^{c)}—^{c)} 9^{c)}HTIB^{c)}DCE^{c)}—^{c)}rt^{c)}—^{c)}44^{c)} 10^{c)}HTIB^{c)}DCE^{c)}—^{c)}80^{c)}—^{c)}55^{c)} 11^{c)}PhIO^{c)}DCE^{c)}S1^{c)}80^{c)}—^{c)}54^{c)} 12^{c)}PhIO^{c)}DCE^{c)}S2^{c)}80^{c)}—^{c)}52^{c)} 13^{c)}PhIO^{c)}DCE^{c)}S3^{c)}80^{c)}—^{c)}63^{c)} 14^{c)}PhIO^{c)}DCE^{c)}S4^{c)}80^{c)}—^{c)}69^{c)} 15^{d)}PhIO^{c)}DCE^{c)}S4^{c)}80^{c)}—^{c)}73^{c)} 16^{d)}PhIO^{c)}DCE^{c)}S4^{c)}80^{c)}—^{c)}70^{c)} 17^{d)}PhIO^{c)}DCE^{c)}S4^{c)}80^{c)}—^{c)}90^{c)} 18^{d)}PhIO^{c)}DCE^{c)}S4^{c)}80^{c)}—^{c)}82^{c)} 19^{d)}PhIO^{c)}DCE^{c)}S4^{c)}80^{c)}—^{c)}77^{c)} 20^{d)}PhIO^{c)}DCE^{c)}S4^{c)}80^{c)}—^{c)}90^{c)}  	Entry ^{a)}	HIR ^{b)}	Solvent ^{c)}	Additive ^{d)}	T (°C) ^{e)}	Yield ^{f)} of 2a (%)	Yield ^{f)} of 3a (%)	1 ^{c)}	PhIO ^{c)}	MeOH ^{c)}	BF ₃ ·OEt ₂ ^{c)}	rt ^{c)}	56 ^{c)}	— ^{c)}	2 ^{c)}	PhIO ^{c)}	MeOH ^{c)}	Et ₃ N ^{c)}	rt ^{c)}	NR ^{c)}	— ^{c)}	3 ^{c)}	PhIO ^{c)}	MeOH ^{c)}	TFA ^{c)}	rt ^{c)}	61 ^{c)}	— ^{c)}	4 ^{c)}	PhIO ^{c)}	MeOH ^{c)}	TfOH ^{c)}	rt ^{c)}	63 ^{c)}	— ^{c)}	5 ^{c)}	PhIO ^{c)}	MeOH ^{c)}	TMSOTf ^{c)}	rt ^{c)}	70 ^{c)}	— ^{c)}	6 ^{c)}	PhIO ^{c)}	MeOH ^{c)}	LiClO ₄ ^{c)}	rt ^{c)}	NR ^{c)}	— ^{c)}	7 ^{c)}	PhIO ^{c)}	MeOH ^{c)}	50% H ₂ SO ₄ ^{c)}	rt ^{c)}	58 ^{c)}	— ^{c)}	8 ^{c)}	PhIO ^{c)}	MeOH ^{c)}	TMSOTf ^{c)}	reflux ^{c)}	81 ^{c)}	— ^{c)}	9 ^{c)}	HTIB ^{c)}	DCE ^{c)}	— ^{c)}	rt ^{c)}	— ^{c)}	44 ^{c)}	10 ^{c)}	HTIB ^{c)}	DCE ^{c)}	— ^{c)}	80 ^{c)}	— ^{c)}	55 ^{c)}	11 ^{c)}	PhIO ^{c)}	DCE ^{c)}	S1 ^{c)}	80 ^{c)}	— ^{c)}	54 ^{c)}	12 ^{c)}	PhIO ^{c)}	DCE ^{c)}	S2 ^{c)}	80 ^{c)}	— ^{c)}	52 ^{c)}	13 ^{c)}	PhIO ^{c)}	DCE ^{c)}	S3 ^{c)}	80 ^{c)}	— ^{c)}	63 ^{c)}	14 ^{c)}	PhIO ^{c)}	DCE ^{c)}	S4 ^{c)}	80 ^{c)}	— ^{c)}	69 ^{c)}	15 ^{d)}	PhIO ^{c)}	DCE ^{c)}	S4 ^{c)}	80 ^{c)}	— ^{c)}	73 ^{c)}	16 ^{d)}	PhIO ^{c)}	DCE ^{c)}	S4 ^{c)}	80 ^{c)}	— ^{c)}	70 ^{c)}	17 ^{d)}	PhIO ^{c)}	DCE ^{c)}	S4 ^{c)}	80 ^{c)}	— ^{c)}	90 ^{c)}	18 ^{d)}	PhIO ^{c)}	DCE ^{c)}	S4 ^{c)}	80 ^{c)}	— ^{c)}	82 ^{c)}	19 ^{d)}	PhIO ^{c)}	DCE ^{c)}	S4 ^{c)}	80 ^{c)}	— ^{c)}	77 ^{c)}	20 ^{d)}	PhIO ^{c)}	DCE ^{c)}	S4 ^{c)}	80 ^{c)}	— ^{c)}	90 ^{c)}
Entry ^{a)}	HIR ^{b)}	Solvent ^{c)}	Additive ^{d)}	T (°C) ^{e)}	Yield ^{f)} of 2a (%)	Yield ^{f)} of 3a (%)																																																																																																																																																																																																																																																																																																		
1 ^{c)}	PhIO ^{c)}	MeOH ^{c)}	BF ₃ ·OEt ₂ ^{c)}	rt ^{c)}	56 ^{c)}	— ^{c)}																																																																																																																																																																																																																																																																																																		
2 ^{c)}	PhIO ^{c)}	MeOH ^{c)}	Et ₃ N ^{c)}	rt ^{c)}	NR ^{c)}	— ^{c)}																																																																																																																																																																																																																																																																																																		
3 ^{c)}	PhIO ^{c)}	MeOH ^{c)}	TFA ^{c)}	rt ^{c)}	61 ^{c)}	— ^{c)}																																																																																																																																																																																																																																																																																																		
4 ^{c)}	PhIO ^{c)}	MeOH ^{c)}	TfOH ^{c)}	rt ^{c)}	63 ^{c)}	— ^{c)}																																																																																																																																																																																																																																																																																																		
5 ^{c)}	PhIO ^{c)}	MeOH ^{c)}	TMSOTf ^{c)}	rt ^{c)}	70 ^{c)}	— ^{c)}																																																																																																																																																																																																																																																																																																		
6 ^{c)}	PhIO ^{c)}	MeOH ^{c)}	LiClO ₄ ^{c)}	rt ^{c)}	NR ^{c)}	— ^{c)}																																																																																																																																																																																																																																																																																																		
7 ^{c)}	PhIO ^{c)}	MeOH ^{c)}	50% H ₂ SO ₄ ^{c)}	rt ^{c)}	58 ^{c)}	— ^{c)}																																																																																																																																																																																																																																																																																																		
8 ^{c)}	PhIO ^{c)}	MeOH ^{c)}	TMSOTf ^{c)}	reflux ^{c)}	81 ^{c)}	— ^{c)}																																																																																																																																																																																																																																																																																																		
9 ^{c)}	HTIB ^{c)}	DCE ^{c)}	— ^{c)}	rt ^{c)}	— ^{c)}	44 ^{c)}																																																																																																																																																																																																																																																																																																		
10 ^{c)}	HTIB ^{c)}	DCE ^{c)}	— ^{c)}	80 ^{c)}	— ^{c)}	55 ^{c)}																																																																																																																																																																																																																																																																																																		
11 ^{c)}	PhIO ^{c)}	DCE ^{c)}	S1 ^{c)}	80 ^{c)}	— ^{c)}	54 ^{c)}																																																																																																																																																																																																																																																																																																		
12 ^{c)}	PhIO ^{c)}	DCE ^{c)}	S2 ^{c)}	80 ^{c)}	— ^{c)}	52 ^{c)}																																																																																																																																																																																																																																																																																																		
13 ^{c)}	PhIO ^{c)}	DCE ^{c)}	S3 ^{c)}	80 ^{c)}	— ^{c)}	63 ^{c)}																																																																																																																																																																																																																																																																																																		
14 ^{c)}	PhIO ^{c)}	DCE ^{c)}	S4 ^{c)}	80 ^{c)}	— ^{c)}	69 ^{c)}																																																																																																																																																																																																																																																																																																		
15 ^{d)}	PhIO ^{c)}	DCE ^{c)}	S4 ^{c)}	80 ^{c)}	— ^{c)}	73 ^{c)}																																																																																																																																																																																																																																																																																																		
16 ^{d)}	PhIO ^{c)}	DCE ^{c)}	S4 ^{c)}	80 ^{c)}	— ^{c)}	70 ^{c)}																																																																																																																																																																																																																																																																																																		
17 ^{d)}	PhIO ^{c)}	DCE ^{c)}	S4 ^{c)}	80 ^{c)}	— ^{c)}	90 ^{c)}																																																																																																																																																																																																																																																																																																		
18 ^{d)}	PhIO ^{c)}	DCE ^{c)}	S4 ^{c)}	80 ^{c)}	— ^{c)}	82 ^{c)}																																																																																																																																																																																																																																																																																																		
19 ^{d)}	PhIO ^{c)}	DCE ^{c)}	S4 ^{c)}	80 ^{c)}	— ^{c)}	77 ^{c)}																																																																																																																																																																																																																																																																																																		
20 ^{d)}	PhIO ^{c)}	DCE ^{c)}	S4 ^{c)}	80 ^{c)}	— ^{c)}	90 ^{c)}																																																																																																																																																																																																																																																																																																		
Entry ^{a)}	HIR ^{b)}	Solvent ^{c)}	Additive ^{d)}	T (°C) ^{e)}	Yield ^{f)} of 2a (%)	Yield ^{f)} of 3a (%)																																																																																																																																																																																																																																																																																																		
1 ^{c)}	PhIO ^{c)}	MeOH ^{c)}	BF ₃ ·OEt ₂ ^{c)}	rt ^{c)}	56 ^{c)}	— ^{c)}																																																																																																																																																																																																																																																																																																		
2 ^{c)}	PhIO ^{c)}	MeOH ^{c)}	Et ₃ N ^{c)}	rt ^{c)}	NR ^{c)}	— ^{c)}																																																																																																																																																																																																																																																																																																		
3 ^{c)}	PhIO ^{c)}	MeOH ^{c)}	TFA ^{c)}	rt ^{c)}	61 ^{c)}	— ^{c)}																																																																																																																																																																																																																																																																																																		
4 ^{c)}	PhIO ^{c)}	MeOH ^{c)}	TfOH ^{c)}	rt ^{c)}	63 ^{c)}	— ^{c)}																																																																																																																																																																																																																																																																																																		
5 ^{c)}	PhIO ^{c)}	MeOH ^{c)}	TMSOTf ^{c)}	rt ^{c)}	70 ^{c)}	— ^{c)}																																																																																																																																																																																																																																																																																																		
6 ^{c)}	PhIO ^{c)}	MeOH ^{c)}	LiClO ₄ ^{c)}	rt ^{c)}	NR ^{c)}	— ^{c)}																																																																																																																																																																																																																																																																																																		
7 ^{c)}	PhIO ^{c)}	MeOH ^{c)}	50% H ₂ SO ₄ ^{c)}	rt ^{c)}	58 ^{c)}	— ^{c)}																																																																																																																																																																																																																																																																																																		
8 ^{c)}	PhIO ^{c)}	MeOH ^{c)}	TMSOTf ^{c)}	reflux ^{c)}	81 ^{c)}	— ^{c)}																																																																																																																																																																																																																																																																																																		
9 ^{c)}	HTIB ^{c)}	DCE ^{c)}	— ^{c)}	rt ^{c)}	— ^{c)}	44 ^{c)}																																																																																																																																																																																																																																																																																																		
10 ^{c)}	HTIB ^{c)}	DCE ^{c)}	— ^{c)}	80 ^{c)}	— ^{c)}	55 ^{c)}																																																																																																																																																																																																																																																																																																		
11 ^{c)}	PhIO ^{c)}	DCE ^{c)}	S1 ^{c)}	80 ^{c)}	— ^{c)}	54 ^{c)}																																																																																																																																																																																																																																																																																																		
12 ^{c)}	PhIO ^{c)}	DCE ^{c)}	S2 ^{c)}	80 ^{c)}	— ^{c)}	52 ^{c)}																																																																																																																																																																																																																																																																																																		
13 ^{c)}	PhIO ^{c)}	DCE ^{c)}	S3 ^{c)}	80 ^{c)}	— ^{c)}	63 ^{c)}																																																																																																																																																																																																																																																																																																		
14 ^{c)}	PhIO ^{c)}	DCE ^{c)}	S4 ^{c)}	80 ^{c)}	— ^{c)}	69 ^{c)}																																																																																																																																																																																																																																																																																																		
15 ^{d)}	PhIO ^{c)}	DCE ^{c)}	S4 ^{c)}	80 ^{c)}	— ^{c)}	73 ^{c)}																																																																																																																																																																																																																																																																																																		
16 ^{d)}	PhIO ^{c)}	DCE ^{c)}	S4 ^{c)}	80 ^{c)}	— ^{c)}	70 ^{c)}																																																																																																																																																																																																																																																																																																		
17 ^{d)}	PhIO ^{c)}	DCE ^{c)}	S4 ^{c)}	80 ^{c)}	— ^{c)}	90 ^{c)}																																																																																																																																																																																																																																																																																																		
18 ^{d)}	PhIO ^{c)}	DCE ^{c)}	S4 ^{c)}	80 ^{c)}	— ^{c)}	82 ^{c)}																																																																																																																																																																																																																																																																																																		
19 ^{d)}	PhIO ^{c)}	DCE ^{c)}	S4 ^{c)}	80 ^{c)}	— ^{c)}	77 ^{c)}																																																																																																																																																																																																																																																																																																		
20 ^{d)}	PhIO ^{c)}	DCE ^{c)}	S4 ^{c)}	80 ^{c)}	— ^{c)}	90 ^{c)}																																																																																																																																																																																																																																																																																																		
Page 2 Right column Paragraph 4	First, a detailed screening of solvents, hypervalent iodine reagents and temperature was carried out (see the SI for complete details).	First, a detailed screening of solvents, hypervalent iodine reagents and temperature was carried out (Supplementary Table S2).																																																																																																																																																																																																																																																																																																						
Page 2 Right column	Next, almost identical results were obtained when using active	Next, almost identical results were obtained when using active hypervalent																																																																																																																																																																																																																																																																																																						

Paragraph 4	hypervalent iodine species formed in situ from iodosobenzene (PhIO) and 4-toluenesulfonic acid (Table 1, entry 11). ^{51, 52}	iodine species formed in situ from iodosobenzene (PhIO) and 4-toluenesulfonic acid (Table 1, entry 11). ^{51, 52}
Page 2 Right column	Table 2. Substrate scope study for synthesis of isoquinolinones 2. ^[a,b]	Fig. 2 Substrate scope study for synthesis of isoquinolinones 2. ^[a] Reaction conditions: 1 (0.5 mmol), PhIO (1.5 equiv) and TMSOTf (0.2 equiv) in MeOH (5.0 mL) reflux for 0.5-2 h. Isolated yield.
Page 2 Right column Paragraph 5	With the optimized conditions in hand, we began to explore the general applicability of the divergent transformation, targeting the N -cyclization by using variously substituted N -phenyl-2-alkenylbenzamides with MeOH as a reaction partner ^{34, 53-57} and solvent (Table 2) being first studied.	With the optimized conditions in hand, we began to explore the general applicability of the divergent transformation, targeting the N -cyclization by using variously substituted N -phenyl-2-alkenylbenzamides with MeOH as a reaction partner ^{34, 53-57} and solvent being first studied (Fig. 2).
Page 3 Left column	Table 3. Substrate scope study for synthesis of iminoisocoumarins 3. ^[a,b]	Fig. 3 Substrate scope study for synthesis of iminoisocoumarins 3. ^[a] Reaction conditions: 1 (0.5 mmol), PhIO (1.5 equiv), 2,4,6-tris-isopropylbenzene sulfonic acid (S4; 1.5 equiv) and LiClO ₄ (1.5 equiv) in DCE (5.0 mL) at 80 °C for 0.1-0.5 h. Isolated yield.
Page 3 Left column Paragraph 6	Next, we came to explore the chemodivergent synthesis of iminoisocoumarins 3 by subjecting substrates 1 to conditions B. As shown in Table 3, o -alkenylbenzamides 1b-v bearing different alkyl substituted R ¹ group and the substituted phenyl ring could smoothly furnish the corresponding iminoisocoumarins 3b-v with sole chemoselectivity in satisfactory to excellent yields (73–92%). Notably, in contrast to the inferior performance of substrate 1g and 1h in N -cyclization mode reaction, iminoisocoumarin product 3g and 3h could be obtained in high yield mediated by the modified Koser's reagent.	Next, we came to explore the chemodivergent synthesis of iminoisocoumarins 3 by subjecting substrates 1 to conditions B (Fig. 3). O -alkenylbenzamides 1b-v bearing different alkyl substituted R ¹ group and the substituted phenyl ring could smoothly furnish the corresponding iminoisocoumarins 3b-v with sole chemoselectivity in satisfactory to excellent yields (73–92%). Notably, in contrast to the inferior performance of substrate 1g and 1h in N -cyclization mode reaction, iminoisocoumarin product 3g and 3h could be obtained in high yield mediated by the modified Koser's reagent.
Page 3 Left column Paragraph 7	In addition, we carried out some control experiments to ascertain the hypervalent iodine species that is responsible for promoting the transformation (Scheme 2). First, we monitored the reaction process of substrate 1a with PhIO in MeOH in the absence of TMSOTf, and it was	In addition, we carried out some control experiments to ascertain the hypervalent iodine species that is responsible for promoting the transformation (Fig. 4). First, we monitored the reaction process of substrate 1a with PhIO in MeOH in the absence of TMSOTf, and it was found that the reaction did not occur

	found that the reaction did not occur (Scheme 2a). Furthermore, the reaction carried out in the absence of PhIO also completely inhibited the transformation of substrate 1a into product 2a. The two results suggested that both hypervalent iodine reagent and Lewis acid are key reagents that enabled the transformation to occur. Next, when LiClO₄ was solely applied to the transformation of 3a, no reaction occurred either. On the basis of this result as well as the outcome of the initial attempt of using HTIB (Table 1, entry 9), we tentatively infer that LiClO₄ could on one hand promote the formation of Koser's reagent, while on the other hand, coordinate with hydroxy group in the hypervalent iodine specie formed in situ.	(Fig. 4a). Furthermore, the reaction carried out in the absence of PhIO also completely inhibited the transformation of substrate 1a into product 2a (Fig. 4b). The two results suggested that both hypervalent iodine reagent and Lewis acid are key reagents that enabled the transformation to occur. Next, when LiClO₄ was solely applied to the transformation of 3a, no reaction occurred either (Fig. 4c). On the basis of this result as well as the outcome of the initial attempt of using HTIB (Table 1, entry 9), we tentatively infer that LiClO₄ could on one hand promote the formation of Koser's reagent, while on the other hand, coordinate with hydroxy group in the hypervalent iodine specie formed in situ.
Page 3 Right column	Scheme 2. Control experiments.	Fig. 4. Control experiments. a Control experiment to verify the necessity of hypervalent iodine reagent and Lewis acid. b Control experiment to verify the effect of LiClO₄.
Page 3 Right column Paragraph 8	To gain insight into the mechanism and chemoselectivity of above systems, we performed density functional theory (DFT) calculations on the reaction of substrate 1a under conditions A and conditions B (Figure 1). As previous work shows, PhI(OMe)₂ is generated in situ by PhIO and MeOH.^{54, 56, 57} It is known that TMSOTf can be present as TMS⁺ + TfO⁻ in organic solvents.^{58, 59} The calculation shows that the complex IM2-1, which is formed by PhI(OMe)₂ and TMSOTf, is thermodynamically favored over reagent 1 by 14.0 kcal/mol (Figure 1a).	To gain insight into the mechanism and chemoselectivity of above systems, we performed density functional theory (DFT) calculations on the reaction of substrate 1a under conditions A and conditions B (Fig. 5). As previous work shows, PhI(OMe)₂ is generated in situ by PhIO and MeOH.^{54, 56, 57} It is known that TMSOTf can be present as TMS⁺ + TfO⁻ in organic solvents.^{58, 59} The calculation shows that the complex IM2-1, which is formed by PhI(OMe)₂ and TMSOTf, is thermodynamically favored over reagent 1 by 14.0 kcal/mol (Fig. 5a).
Page 3 Right column Paragraph 9	For another reaction (Figure 1b), it starts with the coordination of the oxygen to the LiClO₄ which results in the formation of a thermodynamically more stable complex IM3-1 by 10.8 kcal/mol.	For another reaction (Fig. 5b), it starts with the coordination of the oxygen to the LiClO₄ which results in the formation of a thermodynamically more stable complex IM3-1 by 10.8 kcal/mol.
Page 4	Figure 1. DFT-computed potential energy profile for the reaction of 1a under the conditions A and conditions B. Ar = 2,4,6-triisopropyl. (Standard state: 25 °C, 1 mol/L). For details see the Supplementary Information.	Fig. 5 DFT-computed potential energy profile for the reaction of 1a under the conditions A and conditions B. Ar = 2,4,6-triisopropyl. (Standard state: 25 °C, 1 mol/L). (For details see Supplementary data 4)

Page 5 Left column Paragraph 10	Nitrogen or oxygen-attack mode is supported by Hirshfeld charges analysis of IM2-2 and IM3-2, which is capable of predicting electrophilicity and nucleophilicity (Figure 2). By the Hirshfeld charges analysis of IM2-2, we observed that more negative charge (-0.202) was concentrated on the nitrogen atom than that on the oxygen atom (-0.180), indicating that the nitrogen attacking mode is more favorable for IM2-2, thus leading to the formation of N-heterocycle intermediate IM2-3. Furthermore, still by the Hirshfeld charges analysis of IM2-2' shown below, we found that the negative charge at the oxygen atom of the carbonyl group is -0.364, which means that oxygen has higher nucleophilicity than that of nitrogen of IM2-2', this would result in the generation of O-heterocycle intermediate IM2-3'. Thus, N-attack mode is thermodynamically favorable in IM2-2 under reaction condition A. (Figure 2a).⁶⁰	Nitrogen or oxygen-attack mode is supported by Hirshfeld charges analysis of IM2-2 and IM3-2, which is capable of predicting electrophilicity and nucleophilicity (Fig. 6). By the Hirshfeld charges analysis of IM2-2, we observed that more negative charge (-0.202) was concentrated on the nitrogen atom than that on the oxygen atom (-0.180), indicating that the nitrogen attacking mode is more favorable for IM2-2, thus leading to the formation of N-heterocycle intermediate IM2-3. Furthermore, still by the Hirshfeld charges analysis of IM2-2' shown below, we found that the negative charge at the oxygen atom of the carbonyl group is -0.364, which means that oxygen has higher nucleophilicity than that of nitrogen of IM2-2', this would result in the generation of O-heterocycle intermediate IM2-3'. Thus, N-attack mode is thermodynamically favorable in IM2-2 under reaction condition A. (Fig. 6a).⁶⁰
Page 5 Left column	Figure 2. The Hirshfeld charges analysis of IM2-3 and IM3-2.	Fig. 6. The Hirshfeld charges analysis of IM2-3 and IM3-2. Relative free energies and electronic energies are given in kcal/mol.
Page 5 Right column Paragraph 10	Additionally, by the Hirshfeld charges analysis of IM3-2, we observed that more negative charge (-0.309) was concentrated on the oxygen atom of amide group than that on the nitrogen atom (-0.070), which indicates that the O-attack mode is more favorable for IM3-2. And, we respectively calculated the activation energy barrier for O-attack and N-attack in IM3-2. The O-attack pathway needs to overcome an activation energy barrier of 28.1 kcal/mol, while nitrogen atom accomplishes N-attack pathway with a higher barrier of 35.2 kcal/mol. The higher barrier of 35.2 kcal/mol makes it impossible for IM3-2 to undergo an N-attack mode under condition B. The comparison between Hirshfeld charge of the oxygen atom (-0.309) and that of nitrogen atom (-0.070) also illustrate that O-attack mode is more favorable for IM3-2. In summary, O-attack is	Additionally, by the Hirshfeld charges analysis of IM3-2, we observed that more negative charge (-0.309) was concentrated on the oxygen atom of amide group than that on the nitrogen atom (-0.070), which indicates that the O-attack mode is more favorable for IM3-2. And, we respectively calculated the activation energy barrier for O-attack and N-attack in IM3-2. The O-attack pathway needs to overcome an activation energy barrier of 28.1 kcal/mol, while nitrogen atom accomplishes N-attack pathway with a higher barrier of 35.2 kcal/mol. The higher barrier of 35.2 kcal/mol makes it impossible for IM3-2 to undergo an N-attack mode under condition B. The comparison between Hirshfeld charge of the oxygen atom (-0.309) and that of nitrogen atom (-0.070) also illustrate that O-attack mode is more favorable for IM3-2. In summary, O-attack is kinetically favorable under condition B,

	kinetically favorable under condition B, leading to the generation of O -heterocycle intermediate IM3-3 under condition B. (Figure 2b).	leading to the generation of O -heterocycle intermediate IM3-3 under condition B. (Fig. 6b).
Page 5 Right column Paragraph 11	Based on the aforementioned mechanistic studies and the outcomes of the previous reports, ^{36, 52, 54, 61, 62} we postulated a plausible mechanism for the formation of 2a and 3a , shown in Scheme 5. For the formation of product 2a (Scheme 5a), PhI(OMe) ₂ is first generated in situ from the reaction of PhIO with MeOH. Then complex IM2-1 is formed by PhI(OMe) ₂ and TMSOTf, enabling the electrophilic reaction with isomerized substrate 1a' , leading to the formation of iodonium ion IM2-2 . Next, isomerization of the amide occurs via an intermolecular proton shift, and the subsequent nucleophilic attack of the nitrogen atom gives intermediate IM2-4 .	Based on the aforementioned mechanistic studies and the outcomes of the previous reports, ^{36, 52, 54, 61, 62} we postulated a plausible mechanism for the formation of 2a and 3a (Fig. 7). For the formation of product 2a (Fig. 7a), PhI(OMe) ₂ is first generated in situ from the reaction of PhIO with MeOH. Then complex IM2-1 is formed by PhI(OMe) ₂ and TMSOTf, enabling the electrophilic reaction with isomerized substrate 1a' , leading to the formation of iodonium ion IM2-2 . Next, isomerization of the amide occurs via an intermolecular proton shift, and the subsequent nucleophilic attack of the nitrogen atom gives intermediate IM2-4 .
Page 5 Right column	Scheme 3. Possible reaction mechanism.	Fig. 7 Possible reaction mechanism. a Reaction formation mechanism of isoquinolinone 2a . b Reaction formation mechanism of iminoisocoumarin 3a .
Page 6 Left column Paragraph 11	Similarly, with regard to the formation of product 3a (Scheme 5b), the modified Koser's reagent is first formed from the reaction of PhIO with ArSO ₃ H.	Similarly, with regard to the formation of product 3a (Fig. 7b), the modified Koser's reagent is first formed from the reaction of PhIO with ArSO ₃ H.
Page 6 Left column Paragraph 12	In view of the atomic economy of utilizing catalytic hypervalent iodine species, we further investigated the strategy of combining aryl iodine and exogenous oxidant to generate hypervalent iodine species in situ , with a purpose of demonstrating the economic application of this transformation. ⁷⁰⁻⁷⁴	In view of the atomic economy of utilizing catalytic hypervalent iodine species, we further investigated the strategy of combining aryl iodine and exogenous oxidant to generate hypervalent iodine species in situ , with a purpose of demonstrating the economic application of this transformation (Fig. 8). ⁷⁰⁻⁷⁴
Page 6 Left column	Scheme 4. Organocatalytic transformation of chemodivergent synthesis.	Fig. 8 Organocatalytic transformation of chemodivergent synthesis. Experiment to demonstrate the economic application of this transformation.
Page 6 Left column Paragraph 13	In summary, we have presented an exclusive chemodivergent cycloisomerization approach for constructing isoquinolinones and iminoisocoumarins skeletons starting from an identical o -alkenylbenzamides derivative. Notably, the divergent synthesis employed an exclusive 1,2-aryl migration/elimination strategy, which	In summary, we have presented an exclusive chemodivergent cycloisomerization approach for constructing isoquinolinones and iminoisocoumarins skeletons starting from an identical o -alkenylbenzamides derivative. Notably, the divergent synthesis employed an exclusive 1,2-aryl migration/elimination strategy, which is realized by utilizing the

	is realized by utilizing the different hypervalent iodine species generated in situ. In the reaction processes, different hypervalent iodine species was found to play a crucial role in the selectivity switch from N to O-cyclization, with the $\text{PhI}(\text{OMe})_2$ species inducing N-attack and the modified Koser's reagent favoring the O-attack in the cyclization step. DFT studies demonstrated that nitrogen and oxygen atoms of the intermediates in the two reaction systems have different nucleophilicity and thus produce the selectivity of N or O-attack mode.	different hypervalent iodine species generated in situ. In the reaction processes, different hypervalent iodine species was found to play a crucial role in the selectivity switch from N to O-cyclization, with the $\text{PhI}(\text{OMe})_2$ species inducing N-attack and the modified Koser's reagent favoring the O-attack in the cyclization step. DFT studies demonstrated that nitrogen and oxygen atoms of the intermediates in the two reaction systems have different nucleophilicity and thus produce the selectivity of N or O-attack mode.
Page 6 Right column Methods	The combined organic layer was dried over anhydrous Na_2SO_4 and the solvent was removed in vacuo. The residue was purified by flash column chromatography on silica gel to afford target product 2. The combined organic layer was dried over anhydrous Na_2SO_4 and the solvent was removed in vacuo. The residue was purified by flash column chromatography on silica gel to afford target product 3.	The combined organic layer was dried over anhydrous Na_2SO_4 and the solvent was removed in vacuo. The residue was purified by flash column chromatography on silica gel (petroleum ether/ethyl acetate 10:1) to afford target product 2. The combined organic layer was dried over anhydrous Na_2SO_4 and the solvent was removed in vacuo. The residue was purified by flash column chromatography on silica gel (petroleum ether/ethyl acetate 50:1) to afford target product 3.
Page 6 Right column Data Availability	The data that support the findings of this study are available within the article and the Supplementary Information. Details about materials and methods, experimental procedures, characterization data, mechanistic studies, NMR spectra are available in the Supplementary Information. The crystallographic data for compound 2p and 3t can be obtained free of charge from the Cambridge Crystallographic Data Centre (CCDC) under reference number 2201660 (2p) and 2202945 (3t).	All data generated during this study are included in this article and Supplementary Information. Experimental procedure, conditions optimization and product characterization are provided in the Supplementary Information. The NMR spectra of all compounds are available in Supplementary Data 1. The crystallographic data for compound 2p and 3t can be obtained free of charge from the Cambridge Crystallographic Data Centre (CCDC) under deposition numbers 2201660 (2p, Supplementary Data 2) and 2202945 (3t, Supplementary Data 3), respectively. These data can be obtained free of charge from the Cambridge Crystallographic Data Centre via www.ccdc.cam.ac.uk/data_request/cif. DFT calculations are available in Supplementary Data 4.
Page 8 References	54. Zhen, X., Wan, X., Zhang, W., Li, Q., Zhang-Negrerie, D. & Du, Y. Synthesis of Spirooxindoles from N-Arylamide Derivatives via Oxidative $\text{C}(\text{sp}^2)\text{-C}(\text{sp}^3)$ Bond Formation	54. Zhen, X., Wan, X., Zhang, W., Li, Q., Zhang-Negrerie, D. & Du, Y. Synthesis of Spirooxindoles from N-Arylamide Derivatives via Oxidative $\text{C}(\text{sp}^2)\text{-C}(\text{sp}^3)$ Bond Formation

	Mediated by $\text{PhI}(\text{OMe})_2$ Generated in situ . Org. Lett. 21 , 890-894, (2019).	Mediated by $\text{PhI}(\text{OMe})_2$ Generated in situ . Org. Lett. 21 , 890-894, (2019).
Page 8 References	56. Zhang, J., Jalil, A., He, J., Yu, Z., Cheng, Y., Li, G., Du, Y. & Zhao, K. Lactonization with Concomitant 1,2-Aryl Migration and Alkoxylation Mediated by Dialkoxyphenyl Iodides Generated in situ . Chem. Commun. 57 , 7426-7429, (2021).	56. Zhang, J., Jalil, A., He, J., Yu, Z., Cheng, Y., Li, G., Du, Y. & Zhao, K. Lactonization with Concomitant 1,2-Aryl Migration and Alkoxylation Mediated by Dialkoxyphenyl Iodides Generated in situ . Chem. Commun. 57 , 7426-7429, (2021).
Page 8 References	71. Ochiai, M., Takeuchi, Y., Katayama, T., Sueda, T. & Miyamoto, K. Iodobenzene-Catalyzed α -Acetoxylation of Ketones. In Situ Generation of Hypervalent (Diacyloxyiodo)benzenes Using m -Chloroperbenzoic Acid. J. Am. Chem. Soc. 127 , 12244-12245, (2005).	71. Ochiai, M., Takeuchi, Y., Katayama, T., Sueda, T. & Miyamoto, K. Iodobenzene-Catalyzed α -Acetoxylation of Ketones. In Situ Generation of Hypervalent (Diacyloxyiodo)benzenes Using m -Chloroperbenzoic Acid. J. Am. Chem. Soc. 127 , 12244-12245, (2005).
Page 8 Right column Acknowledgements	Y.D. acknowledges the National Natural Science Foundation of China (No. 22071175). We also thank Dr. Jun Xu, Dr. Yan Gao and Prof. Xiangyang Zhang [AIC, SPST/TJU] for providing the analysis support.	YF.D. acknowledges the National Natural Science Foundation of China (No. 22071175). We also thank Dr. Jun Xu, Dr. Yan Gao and Prof. Xiangyang Zhang [AIC, SPST/TJU] for providing the analysis support.
Page 8 Right column Author Contributions	J.H. and Y.D. conceived and designed the experiments. J. H. performed all the experiments, prepared the manuscript and supporting information. F.-H.,D. performed all the DFT calculations work and collated the calculated data. C.Z. and Y.D. directed the research and revised the manuscript.	JX.H. , and YF.D. conceived and designed the experiments. JX. H. performed all the experiments, prepared the manuscript and supporting information. F-H.D. performed all the DFT calculations work and collated the calculated data. C.Z. , and YF.D. directed the research and revised the manuscript.

The end